# Pancreatic glycoprotein 2 is a first line of defense for mucosal protection in intestinal inflammation

Yosuke Kurashima[1,2,3,4,5,18]✉, Takaaki Kigoshi [1,6,18], Sayuri Murasaki[1,3], Fujimi Arai[1,3], Kaoru Shimada[1,3], Natsumi Seki[7,8], Yun-Gi Kim[8], Koji Hase [7,3], Hiroshi Ohno [9,10,11], Kazuya Kawano[9], Hiroshi Ashida[12,13], Toshihiko Suzuki[12], Masako Morimoto[2], Yukari Saito[2], Ai Sasou[1,3], Yuki Goda[1,3], Yoshikazu Yuki[1,3], Yutaka Inagaki [14], Hideki Iijima[15], Wataru Suda[16], Masahira Hattori[16,17] & Hiroshi Kiyono [1,3,4,5]✉

Increases in adhesive and invasive commensal bacteria, such as *Escherichia coli*, and sub-sequent disruption of the epithelial barrier is implicated in the pathogenesis of inflammatory bowel disease (IBD). However, the protective systems against such barrier disruption are not fully understood. Here, we show that secretion of luminal glycoprotein 2 (GP2) from pan-creatic acinar cells is induced in a TNF–dependent manner in mice with chemically induced colitis. Fecal GP2 concentration is also increased in Crohn's diease patients. Furthermore, pancreas-specific GP2-deficient colitis mice have more severe intestinal inflammation and a larger mucosal *E. coli* population than do intact mice, indicating that digestive-tract GP2 binds commensal *E. coli*, preventing epithelial attachment and penetration. Thus, the pancreas–intestinal barrier axis and pancreatic GP2 are important as a first line of defense against adhesive and invasive commensal bacteria during intestinal inflammation.

[1] Department of Mucosal Immunology, The University of Tokyo Distinguished Professor Unit, The Institute of Medical Science, The University of Tokyo, Tokyo, Japan. [2] Department of Innovative Medicine, Graduate School of Medicine, Chiba University, Chiba, Japan. [3] International Research and Development Center for Mucosal Vaccines, The Institute of Medical Science, The University of Tokyo, Tokyo, Japan. [4] Division of Gastroenterology, Department of Medicine, CU-UCSD Center for Mucosal Immunology, Allergy and Vaccines, University of California, San Diego, CA, USA. [5] Mucosal Immunology and Allergy Therapeutics, Institute for Global Prominent Research, Graduate School of Medicine, Chiba University, Chiba, Japan. [6] Department of Pediatrics, Graduate School of Medicine, Tohoku University, Miyagi, Japan. [7] Division of Biochemistry, Graduate School of Pharmacological Science, Keio University, Tokyo, Japan. [8] Research Center for Drug Discovery, Faculty of Pharmacy, Keio University, Tokyo, Japan. [9] Laboratory for Intestinal Ecosystem, RIKEN Center for Integrative Medical Sciences, Kanagawa, Japan. [10] Division of Immunobiology, Department of Medical Life Science, Graduate School of Medical Life Science, Yokohama City University, Kanagawa, Japan. [11] Intestinal Microbiota Project, Kanagawa Institute of Industrial Science and Technology, Kanagawa, Japan. [12] Department of Bacterial Infection and Host Response, Graduate School of Medical and Dental Sciences, Tokyo Medical and Dental University, Tokyo, Japan. [13] Medical Mycology Research Center, Chiba University, Chiba, Japan. [14] Center for Matrix Biology and Medicine, Graduate School of Medicine, Tokai University, Kanagawa, Japan. [15] Department of Gastroenterology and Hepatology, Graduate School of Medicine, Osaka University, Osaka, Japan. [16] Laboratory for Microbiome Sciences, RIKEN Center for Integrative Medical Sciences, Kanagawa, Japan. [17] Graduate School of Advanced Science and Engineering, Waseda University, Tokyo, Japan. [18]These authors contributed equally: Yosuke Kurashima, Takaaki Kigoshi. ✉email: yosukek@chiba-u.jp; hkiyono@health.ucsd.edu

nflammatory bowel diseases, such as Crohn's disease (CD) and ulcerative colitis (UC), are associated with dysregulation of the intestinal mucosal barrier and dysbiosis[1–3]. Recently, it has been reported that pancreatic autoantibodies against a major pancreatic glycoprotein of the zymogen granule membrane, glycoprotein 2 (GP2), and CUB and zona pellucida-like domain-containing protein 1 are associated with the severity of inflammation in patients with CD[4–6]. It is also considered that breakdown of tolerance against those pancreatic proteins causes induction of self-antibodies[7]. Patients with pancreatitis show higher susceptibility to CD[8]. Also, several lines of evidence show a high prevalence of mucosal-associated commensal bacteria, including *Escherichia coli*, particularly adherent-invasive *E. coli*, in CD patients and that these bacteria play a key role in the pathogenesis of CD[9]. It is reported that adherent-invasive *E. coli* colonization during acute infectious gastroenteritis is a major risk for CD onset[10–12]. Adherent-invasive *E. coli* are resistant to clearance by the immune system, such as by macrophages, and induce markedly increased expression of the inflammatory cytokine tumor necrosis factor-alpha (TNF) by bacteria infected macrophages[13]. TNF plays a pivotal role in the induction and amplification of the inflammatory cascade in CD[14].

GP2 is a glycosylphosphatidylinositol (GPI)-anchored protein that was first reported in pancreas[15]. GP2 is the most abundant pancreatic membrane protein of zymogen granules, which are produced by pancreatic acinar cells and contain various kinds of digestive enzymes[16]. A GPI-anchor-cleaved form of GP2 is shed into pancreatic secretions[17]. In our previous study, we found that microfold cells (M cells) in follicle-associated epithelium of Peyer's patches also express GP2, which acts as a transcytotic receptor for type 1 fimbrial adhesin (FimH), a component of type 1 pili, expressed by *E. coli* and *Salmonella typhimurium*[18–20]. Transcytosis through M cells delivers antigens to the underlying dendritic cells and initiates antigen-specific mucosal immune responses[21]. It has been suggested that anti-GP2 antibodies neutralize the function of pancreatic GP2 or GP2 expressed on M cells in Peyer's patches[19,22]; however, the role of pancreatic GP2 in intestinal inflammation and homeostasis remains to be clarified.

A GP2 homolog, Tamm–Horsfall protein, which is the most abundant protein in mammalian urine, is reported to have anti-bacterial effects in the urinary tract[23,24]. Tamm–Horsfall protein binds to FimH expressed by uropathogenic *E. coli* and inhibits these bacteria from attaching to and invading the urothelial epithelium[25]. It has been reported that the amount of GP2 on the surface of intestinal bacteria is decreased in patients with CD[26]. Thus, we hypothesized that intestinal GP2 may have similar protective effects for the intestinal epithelium as does the Tamm–Horsfall protein in the urothelial epithelium.

In this study, we identify a biological function of pancreatic GP2 in the harsh environment of the intestinal tract, controlling bacterial attachment and penetration for intestinal homeostasis.

## Results

**GP2 distribution in the digestive system**. In addition to our previous report that GP2 is expressed on the apical surface of intestinal M cells[18], we found dissemination of GP2 at the luminal side of the intestine (Supplementary Fig. 1). Building on these findings, we examined the precise distribution of GP2 in the digestive system, including the stomach, pancreas, small intestine, and colon, by means of immunohistochemistry (Fig. 1a). GP2 was detected in the pancreas but not in the luminal compartment of the stomach, which is consistent with a previous report that GP2 is expressed by pancreatic acinar cells[15]. In addition, GP2 was detected throughout the lumen of the small intestine and colon,

indicating that GP2 secreted by the pancreas diffuses throughout the intestinal lumen. It has been reported that the salivary glands secret GP2[27]; however, we did not detect GP2 expression in either the sublingual or submandibular areas of an intact mouse, further indicating that intestinal GP2 originates from the pancreas (Supplementary Fig. 2). Because the intestinal and colonic mucosae are coated and protected by a mucus layer, we counterstained sections of intact wild-type (WT) mouse colon for mucin 2 (MUC2)[28] and found that it was co-localized with GP2 (Fig. 1b). We confirmed the specificity of the GP2 signal by histological analysis of GP2-deficient and WT mice using an isotype control and by western blotting (Fig. 1b and Supplementary Fig. 3a, b). We also found that GP2 was present in the outer mucin layer of the colon (Supplementary Fig. 4).

The number of GP2-expressing epithelial cells (i.e., M cells) is reportedly increased by bacterial stimulation[29]. We therefore examined specific pathogen-free, germ-free, and antibiotic-treated mice and found GP2 in the luminal content of all three types of mouse, suggesting that GP2 is constitutively secreted from the pancreas irrespective of the presence of intestinal bacterial (Supplementary Fig. 5).

**Increase of GP2 production in colitis**. Next, to examine GP2 expression in colitis, we compared gene expressions between intact mice and mice in which colitis was induced by administration of dextran sodium sulfate (DSS). In the mice with DSS-colitis, no significant alteration of pathological, histological (e.g., inflammatory cell accumulation), or physiological (e.g., expression of pancreatic lipase) conditions were found in the pancreas (Fig. 2a, b). Intriguingly, significantly elevated expression of pancreatic *Gp2* gene was observed; however, no significant increase was found in the follicular-associated epithelium[30] of Peyer's patches or colonic cellular populations, including collagen-expressing mesenchymal cells, EpCAM+ epithelial cells, and CD45+ hematopoietic cells (Fig. 2c). These results indicated that the production of GP2 was elevated in pancreas but not in colonic cell populations (Fig. 2c). Indeed, increased GP2 expression in pancreas was confirmed by immunohistochemical analysis, implicating the involvement of inter-organ communication signals between pancreas and colon during intestinal inflammation (Fig. 2d). In addition, the concentrations of GP2 in pancreatic juice and luminal washout were increased in DSS-colitis mice compared with intact mice, indicating increased GP2 production under the inflammatory condition (Fig. 2e). Furthermore, the concentration of GP2 in the stool of patients with inflammatory bowel disease, especially in that of CD patients, also showed increased concentrations of GP2 compared with that in healthy individuals (Fig. 2f). Thus, the findings generated by the mouse study were considered to accurately reflect the human intestinal inflammatory condition.

The increased amounts of GP2 in the pancreatic juice and intestinal lumen in colitis mice (Fig. 2e) suggested increased granular release of GP2 from pancreatic acinar cells. Granular secretion from acinar cells is known to be mediated by the series of proteins involved in the transport, mobility, tethering and membrane docking, and fusion[31]. To examine whether the granular transport system is enhanced in intestinal inflammation, we determined the expressions of the genes encoding several GTPases (Rab3b, Rab6a, Rab8a, Rab27b, and Rap1a)[32] and vesicular trafficking proteins (vesicle associated membrane protein [Vamp] 2, 3, and 8 and syntaxin 3 and 7)[33–35] that are responsible for the transport and mobility of granular vesicles in acinar cells. We found no increases in any of the gene expressions (Supplementary Fig. 6a), indicating that the granular transport system of pancreatic acinar cells is unaltered during intestinal inflammation. Next, we examined the expressions of the genes

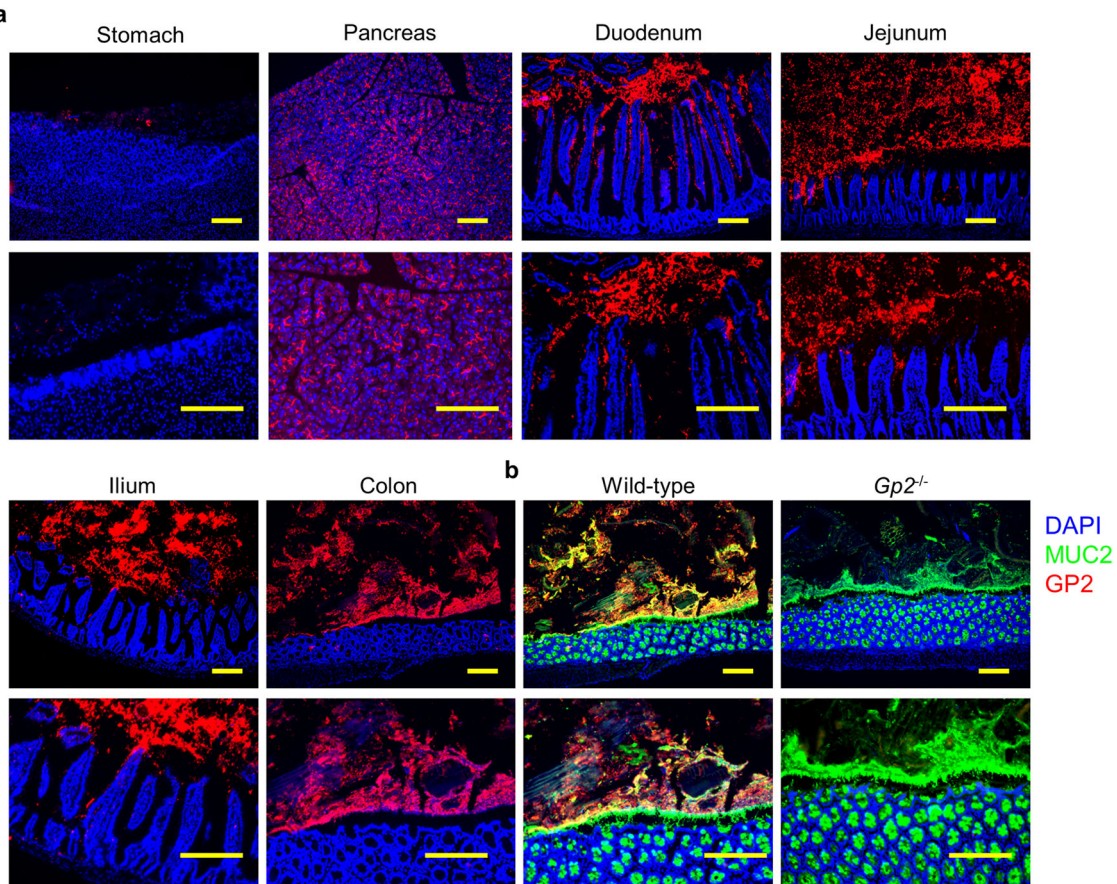

**Fig. 1 GP2 in mouse digestive tract. a** Immunohistochemical analysis of GP2 distribution in the gastrointestinal tract (stomach, duodenum, jejunum, ileum, and colon) with luminal contents and pancreas of an intact mouse; labels correspond with those shown in. Representative data of five independent experiments are shown. GP2 (red), DAPI (blue). Scale bars, 100 μm. **b** Counter staining of mice colon with MUC2 (green) of WT and Gp2-deficient mice are shown. Representative data of 3 independent experiments are shown. Scale bars, 100 μm.

that encode several trypsins, phospholipases, and carboxypeptidases that cleave the GPI anchor of GP2 and promote membrane release of GP2 into secretory granules[36,37] (Supplementary Fig. 6b). Although pancreatic acinar cells are known to express phospholipases (e.g., PLCδ) and carboxypeptidases (e.g., CPA2)[36], we did not find increased expressions of those enzymes, suggesting that it is an increase in the de novo production of GP2 by acinar cells that leads to the elevation of luminal GP2 (Fig. 2c–e, Supplementary Fig. 6b).

Based on these findings, we hypothesized that the pancreas is an important protective organ against intestinal inflammation and that it exerts its protective effect by increasing GP2 production via an inflammation-mediated pancreas–colon axis. We examined this idea further by determining the expressions of genes encoding four antimicrobial proteins (regenerating islet-derived [Reg] family protein 2, Reg3α, Reg3β, and Reg3γ) in the pancreas of DSS-colitis mice[38]. The expressions of *Reg2*, *Reg3b*, and *Reg3g* tended to be increased in the DSS-colitis mice (Supplementary Fig. 7), suggesting the existence of inter-organ communication signals between the pancreas and colon during intestinal inflammation.

**Induction of pancreatic GP2 by inflammatory cytokines.** Given our present data showing elevated production of GP2 by pancreatic acinar cells during colitis (Fig. 2c–e), we next examined the pancreas–colon axis and the mechanisms underlying how colonic inflammation leads to increased synthesis of pancreatic GP2. First, we constructed an in vitro acinar cell culture system

from mice pancreas (Supplementary Fig. 8)[39,40]. To confirm if acinar cells were appropriately cultured, we generated *Pancreas transcription factor 1 subunit alpha* (*Ptf1a*)$^{Cre-ERTM}$-tdTomato mice, which were given tamoxifen to induce recombination of Cre-ERTM and confirmed the Tomato signal in PTF1A-expressing cell populations, especially pancreatic acinar cells[41]. Next, we collected and cultured pancreatic acinar cells from the *Ptf1a*$^{Cre-ERTM}$-tdTomato mice (Fig. 3a). The cells were treated with 4-hydroxytamoxifen (4OHT) and red fluorescence was confirmed, as seen in vivo (Fig. 3b, c). These in vivo and in vitro analyses revealed that our in vitro acinar cell culture method was suitable for use in the subsequent experiment.

The pancreas was isolated from *Ptf1a*$^{Cre-ERTM}$-tdTomato mice and cultured with a colitis-associated inflammatory stimulant (i.e., interleukin [IL]-6 or TNF), bacterial endotoxin (i.e., *E. coli* lipopolysaccharide [LPS]), or a GP2-inducing cytokine for M cells (i.e., receptor activator of nuclear factor-κB ligand: RANKL) (Fig. 3d)[14,42,43]. TNF, but none of the other stimulants, upregulated *Gp2* expression in pancreatic acinar cells (Fig. 3d). Tissue cultivation has confirmed bacterial translocation to the mesenteric lymph nodes during colitis in mice[44]. We therefore examined whether bacteria also translocate to the pancreas and whether direct bacterial stimulation of acinar cells occurs during gut inflammation. We cultured digested pancreas from DSS-treated mice on bacterial plates; however, we did not find any translocated bacteria in the pancreas (Supplementary Fig. 9), suggesting that gut bacteria do not directly stimulate acinar cells to produce GP2.

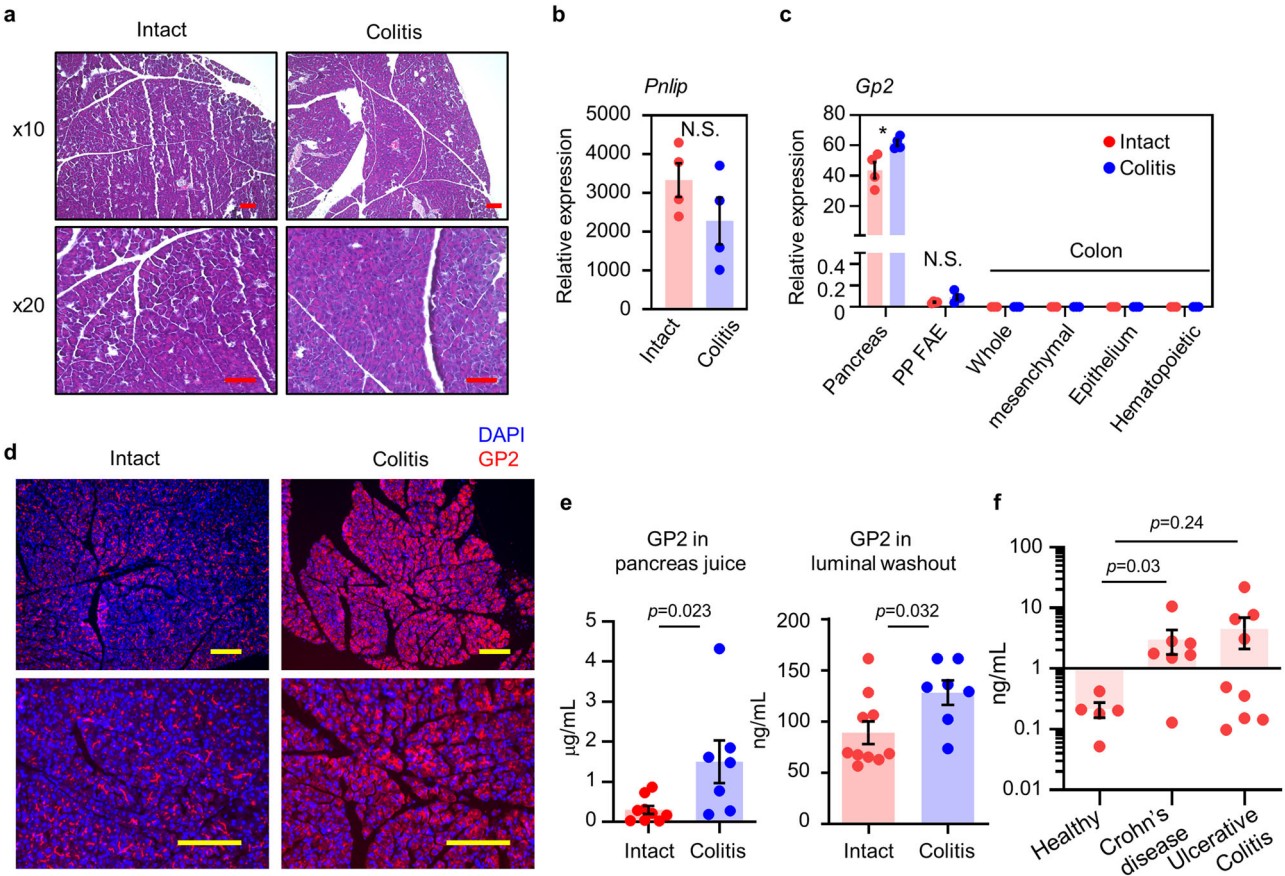

**Fig. 2 Upregulation of pancreatic GP2 production in colitis. a** Hematoxylin and eosin staining of pancreas in intact or colitis mice are shown. Scale bars, 100 μm. Data are representative of at least three independent experiments. Gene expression of *Pnlip* in pancreas (**b**) and of *Gp2* in the indicated tissue and cellular components of the gastrointestinal tract (**c**) in intact mice and mice with acute colitis (*n* = 4/group, two-tailed unpaired *t*-test). **p* = 0.021. Data are presented as mean values ± SEM. N.S. indicates not significant. **d** GP2 expression in pancreas. Representative data of four independent experiments are shown. GP2 (red), DAPI (blue). Scale bars, 100 μm. **e** Concentrations of secretory GP2 in pancreatic juice and intestinal lumen, as measured by enzyme-linked immunosorbent assay (ELISA) (intact, *n* = 10; colitis, *n* = 7, two-tailed Mann–Whitney *U* test). Data are presented as mean values ± SEM. **f** Fecal GP2 concentration was determined by ELISA and compared between healthy people (*n* = 5), CD patients (*n* = 7), and ulcerative colitis patients (*n* = 9), two-tailed Mann–Whitney *U* test. Data are presented as mean values ± SEM. Source data are provided as a Source Data file.

Next, to examine the role of TNF in the induction of GP2 production, we systemically administered TNF to WT mice and found increased production of GP2 in pancreatic juice (Fig. 3e, f). A previous study has shown elevated levels of TNF in the serum and mesenteric fat of colitis mice[44]; we likewise found increased levels of TNF in the blood circulation of our DSS-colitis mice (Fig. 3g). We also examined the concentrations of TNF in the organ matrices surrounding the pancreas (i.e., omentum, mesenteric fat, retro-peritoneal fat, and epididymal fat; Fig. 3h) and found increased tissue concentrations of TNF in the pancreas and mesenteric fat of DSS-colitis mice compared with those in intact mice (Fig. 3i). These findings suggest that systemic TNF that is accumulated in the pancreas and TNF that is derived from the abdominal-organ matrix act as inflammatory messengers that stimulate pancreatic GP2 production. To further examine the involvement of TNF in GP2 production, we administrated TNF-neutralizing or control anti-bodies to DSS-colitis mice during the colitis induction period. Compared with control antibody administration, TNF-neutralizing antibody treatment reduced GP2 production in the pancreas of the mice (Fig. 3j), providing supportive evidence for the role of TNF in the induction of GP2 production in the pancreas.

**Role of GP2 for the control of commensal bacteria invasion in colitis.** To examine the importance of pancreatic GP2 in the control of host–microbial interaction in intestinal inflammation, we first examined whether *Gp2*[−/−] mice have obvious mucosal immunodeficiency because GP2 is expressed in M cells, which are gatekeepers of the gut mucosal immune system[18]. We measured fecal total IgA concentrations because ablation of M cells reduced fecal IgA[45] and thus is considered as an index of mucosal immunodeficiency; however, our current result showed that fecal total IgA was comparable between the two mouse groups (Fig. 4a).

Although *Gp2*[−/−] mice do not show any signs of nutrient malabsorption or diarrhea[46], the lack of GP2 might have an effect on the commensal bacteria, which continuously interact with the gut mucosal immune system. To examine this idea further, we compared the composition of the commensal microbiota of WT and *Gp2*[−/−] mice by mean of 16S rRNA gene sequencing analysis; no significant differences were observed at the phylum or genus levels (Fig. 4b and Supplementary Fig. 10a). An in-depth analysis of the microbiota in *Gp2*[−/−] mice revealed that the populations of *Helicobacter* sp. *MIT 01-6451* (*p* = 0.034, two-tailed unpaired *t*-test) and *Clostridium* sp. *ASF356* (*p* = 0.037, two-tailed unpaired *t*-test) were increased, whereas that of *Bacteroides acidifaciens* (*p* = 0.040, two-tailed unpaired *t*-test) was reduced (Supplementary Fig. 10b); however, the association between these species and the severity of colitis is yet to be examined in

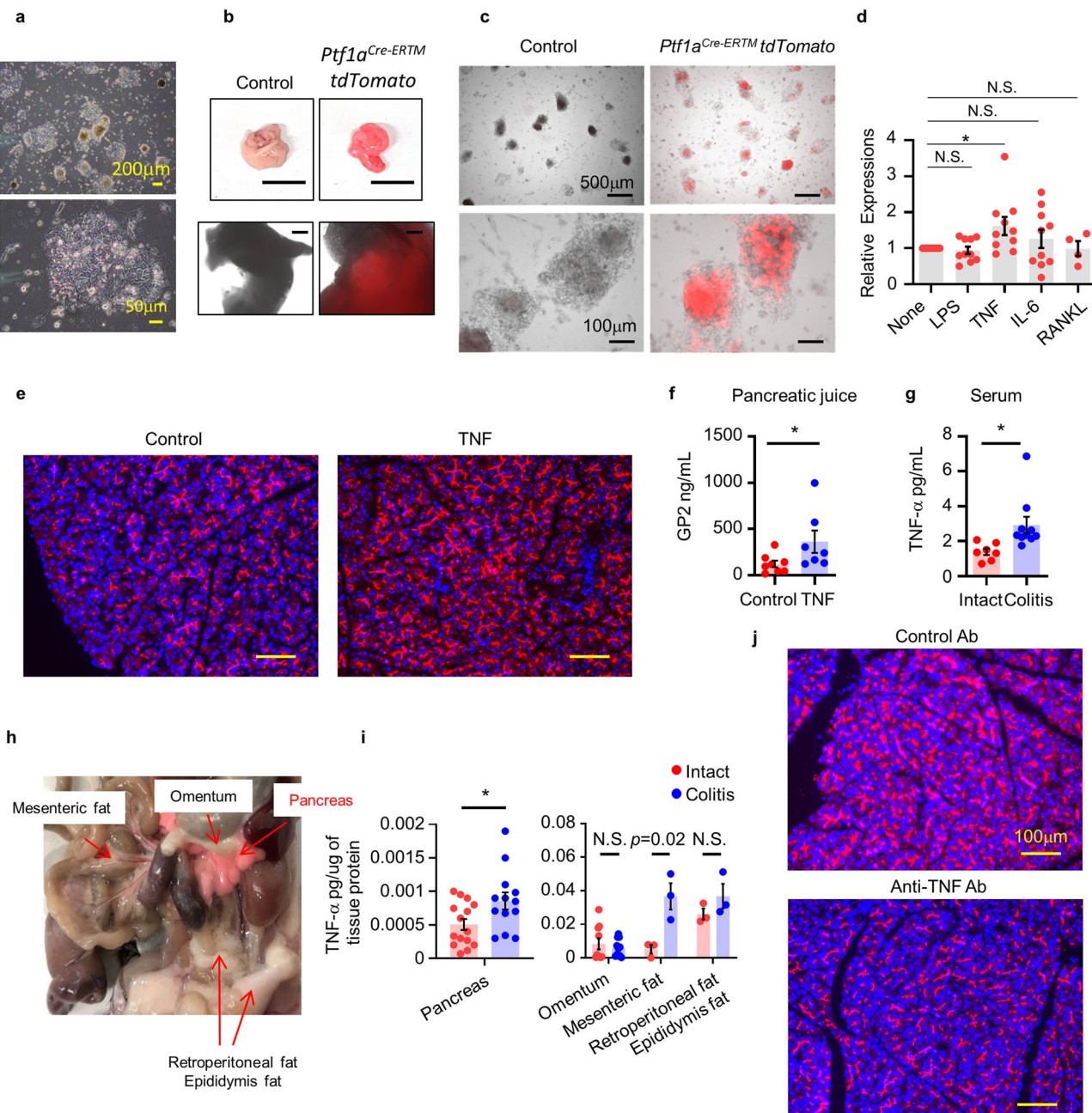

**Fig. 3 Inflammatory signal upregulates *Gp2* expression in pancreatic acinar cells. a** Cultured pancreatic acinar cells are shown. **b** Pancreas of control (upper) or *Ptf1a^{Cre-ERTM}*Cre-tdTomato (bottom) mice received tamoxifen are shown. **c** Cultured pancreatic acini from control of *Ptf1a^{Cre-ERTM}-tdTomato* mice stimulated with 4OHT are shown. Data are representing for three independent experiments. **d** *Gp2* expression in no stimulation (None) or stimulated by 200 ng/mL LPS, 20 ng/mL TNF, 20 ng/mL IL-6 ($n = 10$ /group), 100 ng/mL RANKL ($n = 4$) pancreatic acini were examined. Relative expressions mRNA levels were determined by quantitative RT-PCR and normalized to *Gapdh*. *$p = 0.040$ (one-way ANOVA). Data are presented as mean values ± SEM., N.S. indicates not significant. **e** Pancreas of mice systemically administered PBS (control) or TNF (100 ng, two times). GP2 (red), DAPI (blue). Representative data of three independent experiments are shown. **f** GP2 concentration in pancreatic juice. *$p = 0.04$ (two-tailed Mann–Whitney U test). Data are presented as mean values ± SEM. **g** TNF concentration in serum *$p = 0.021$ (two-tailed unpaired t-test). Data are presented as mean values ± SEM. **h** Representative photograph showing the abdominal-organ matrices surrounding the pancreas in *Ptf1a^{Cre-ERTM}-tdTomato* mice that received tamoxifen. **i** TNF concentration in abdominal-organ matrices, as determined by enzyme-linked immunosorbent assay. Pancreas (intact $n = 15$, colitis $n = 13$), Omentum (intact $n = 10$, colitis $n = 11$) mesenteric fat (intact $n = 15$, colitis $n = 13$), Retroperitoneal fat and Epididymis fat (intact $n = 3$, colitis $n = 3$) *$p < 0.05$ (two-tailed unpaired t-test). Data are presented as mean values ± SEM. **j** 100 μg of anti-TNF antibody (Biolegend, MP6-XT22, #506332) or Isotype Rat IgG2a (Biolegend, RTK2758, #400503) was intraperitoneal administered three times (on days 1, 3, and 5) during DSS treatment, and pancreases were collected on day 6. GP2 (red), DAPI (blue). Representative data of three independent experiments are shown. Source data are provided as a Source Data file.

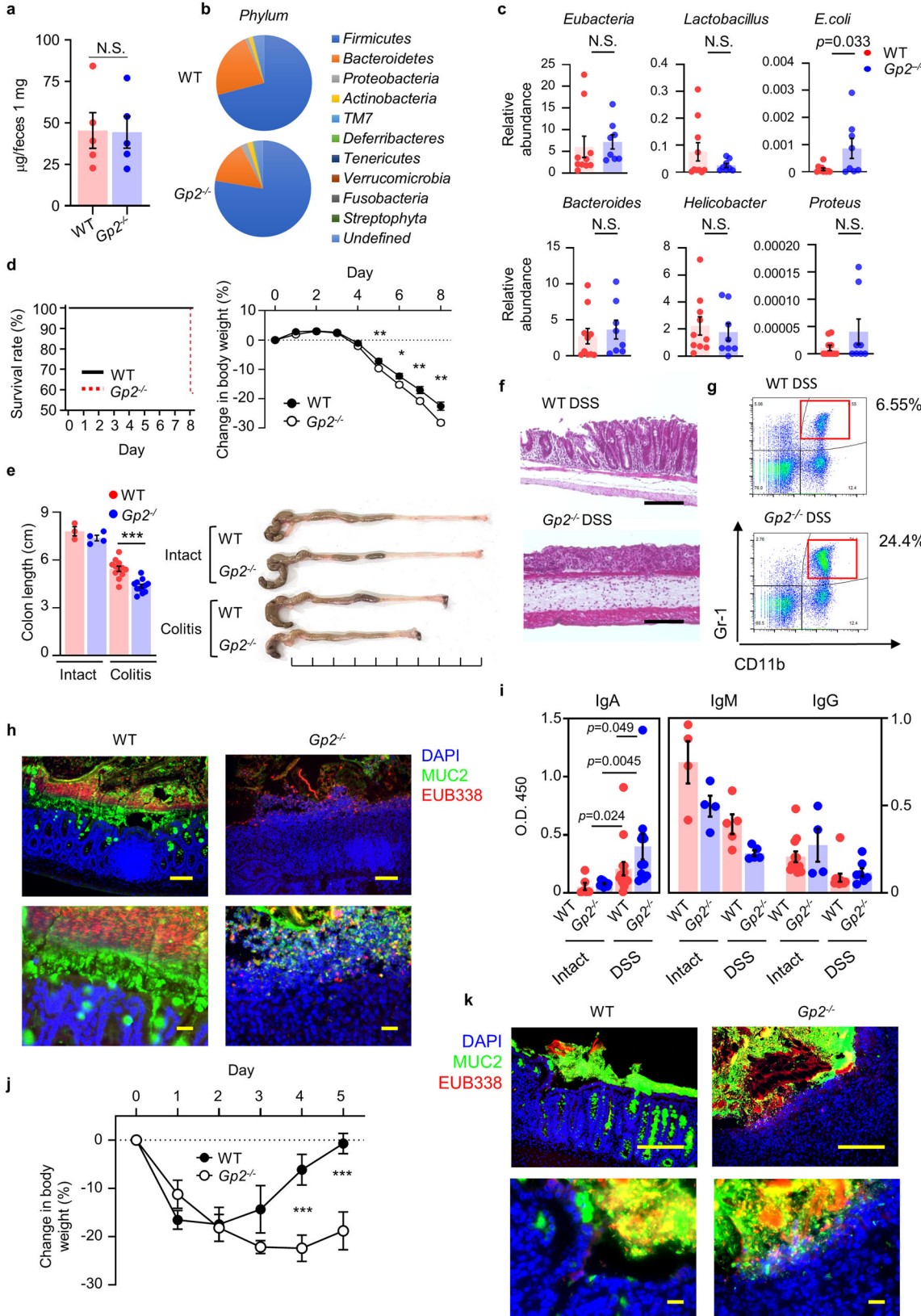

detail[47,48]. To further determine whether GP2 has any effect on the diversity of the luminal microbiota, we examined how deletion of GP2 expression at a young age (i.e., 2 weeks of age) affected the luminal microbiota of *Ptf1a*[Cre-ERTM] *Gp2*[flox/flox] (*GP2*[Panc]) mice in which pancreatic GP2 is specifically deleted by administration of tamoxifen. We did not find any significant differences in the luminal microbiota of the GP2-deficient mice compared with that of control mice (Supplementary Fig. 11). Together, these results indicate that GP2 deficiency does not influence the luminal microbiota.

In addition to luminal dysbiosis, increased numbers of mucosal bacteria is also a risk factor for the development of inflammatory

**Fig. 4 Deletion of systemic GP2 results in severe colitis. a** Fecal total IgA concentration in WT and $Gp2^{-/-}$ mice, as determined by ELISA ($n = 5$). N.S. not significant (two-tailed unpaired $t$-test). Data are presented as mean values ± SEM. **b** 16S rRNA gene sequencing of fecal bacteria. *Phylum* levels of bacteria are shown. **c** qPCR analysis of mucosal bacteria from WT ($n = 9$) and $Gp2^{-/-}$ ($n = 8$) mice were examined. *$p < 0.05$., N.S. indicates not significant (two-tailed unpaired $t$-test). Data are presented as mean values ± SEM. **d** Survival ratio and body weight change of WT and $Gp2^{-/-}$ mice with acute colitis ($n = 12$/group). *$p < 0.05$, **$p < 0.01$ (two-tailed unpaired $t$-test). Data are presented as mean values ± SEM. **e** Colon length and representative pictures of colon are shown, intact; WT ($n = 3$), KO ($n = 4$), and colitis; WT, KO ($n = 12$). ***$p < 0.0001$ (two-tailed unpaired $t$-test). Data are presented as mean values ± SEM. **f** Hematoxylin and eosin staining of colon at day 8 of DSS treatment are shown. Scale bars: 100 μm. Data are representative of three independent experiments. **g** Flow cytometry analysis of infiltrated neutrophils is shown. Representative data are shown. **h** Immunohistochemical analysis of colitis (DSS 2%, day 8) colon with luminal contents of WT and $Gp2^{-/-}$ mice. Bacteria (red), MUC2 (green), DAPI (blue). Scale bars: upper panels, 100 μm; lower panels, 20 μm. Data are representative of three independent experiments. **i** Serum anti-*E. coli* IgM, intact ($n = 4$), and colitis ($n = 5$), IgA, intact WT ($n = 6$), KO ($n = 10$), and colitis WT ($n = 15$), KO ($n = 11$), IgG intact WT ($n = 12$), KO ($n = 4$), and colitis WT ($n = 11$), KO ($n = 7$) in intact and colitis WT and $Gp2^{-/-}$ mice were analyzed by ELISA (Kruskal-Wallis test followed by Mann–Whitney $U$ test). Data are presented as mean values ± SEM. **j** Body weight changes after the induction of acute TNBS-induced colitis in WT and $Gp2^{-/-}$ mice. ***$p < 0.01$ (two-tailed unpaired $t$-test). Data are presented as mean values ± SEM. **k** Immunohistochemical analysis of colon and luminal contents during TNBS-induced colitis in $Gp2^{Panc}$ and control mice. EUB338 bacteria (red), MUC2 (green), DAPI (blue). Scale bars: upper panels, 100 μm; lower panels, 20 μm. Data are representative of two independent experiments. Source data are provided as a Source Data file.

bowel disease[49]. We therefore examined whether GP2 deficiency alters the mucosal bacterial load in WT and GP2-deficient mice (Fig. 4c). After removal of the colonic contents, samples of the mucosal bacteria were collected by scraping the colonic mucosa. Then, the sizes of the total bacterial population (*Eubacteria*) and of several individual bacterial populations (e.g., *E. coli*, *Lactobacillus*, *Bacteroides*, *Helicobacter*, and *Proteus*) were determined by qPCR. Among the gastrointestinal bacteria species, the *E. coli* population was increased in the GP2-deficient mice compared with in the WT mice, indicating that GP2 contributes to the control of the mucosal microbiota in the intestine via the *E. coli* population and therefore that GP2-deficiency might increase the severity of colitis (Fig. 4c).

To further examine the involvement of GP2 in colitis, colitis was induced in GP2-deficient mice, and the survival rate was determined as an index of the severity of inflammation. The survival rate on day 8 was reduced in $Gp2^{-/-}$ mice compared with WT mice ($n = 12$, $p < 0.05$, two-tailed log-rank test; Fig. 4d). In addition, body weight was significantly decreased (Fig. 4d) and colon length was shortened in $Gp2^{-/-}$ mice compared with in WT mice (Fig. 4e). Histological analysis as well as flow cytometry analysis revealed that marked epithelial damage and neutrophil infiltration were observed in $Gp2^{-/-}$ mice (Fig. 4f,g). We next conducted a fluorescent in situ hybridization analysis of bacteria by using a pan marker of bacteria (EUB338) and found that the amount of adhesive or invasive bacteria (as EUB338+ red fluorescent cells) was markedly greater in $Gp2^{-/-}$ mice compared with in WT mice (Fig. 4h). These results indicated that intestinal GP2 protect from bacterial infiltration into the intestinal mucosa to prevent acceleration of inflammation. Indeed, serum Ig levels against *E. coli* revealed that among Ig class, especially serum IgA against *E. coli* was significantly increased in $Gp2^{-/-}$ colitis mice, indicating that immunological responses to infiltrated or disseminated *E. coli* occurred in the systemic compartment (Fig. 4i).

We also examined the severity of colitis in a 2,4,6-trinitrobenzene sulfonic acid (TNBS)-induced colitis model[50] using WT and $Gp2^{-/-}$ mice, and the results were consistent with the DSS-induced colitis data (Fig. 4d–h); body weight and colon length analyses showed severe disease in $Gp2^{-/-}$ mice compared with in WT mice (Fig. 4j and Supplementary Fig. 12a). In addition, histological and flow cytometry analyses revealed marked neutrophil infiltration and bacterial invasion in the TNBS-colitis $Gp2^{-/-}$ mice compared with in the TNBS-colitis WT mice (Fig. 4k and Supplementary Fig. 12b, c). Importantly, as in the DSS-colitis model (Fig. 4c–g), the production of pancreatic GP2 was markedly upregulated in the TNBS-colitis WT mice (Supplementary Fig. 12d). These findings indicate that GP2 in

the digestive tract binds to commensal *E. coli* and regulates the bacterial population by preventing the bacteria from attaching to and penetrating the epithelium during intestinal inflammation. Thus, the reduction of GP2 expression and/or neutralization of its function may increase the risk of developing colitis.

**Pancreatic acinar cells, not M cells, are the source of intestinal GP2.** Because GP2 is expressed both by pancreatic acinar cells and M cells, we next generated conditional GP2-deficient mice to examine the roles of these cells in GP2 production. In this experiment, we targeted Villin 1 (VIL1), which is expressed mainly by intestinal epithelial cells, including M cells[51]. It has also been reported that VIL1 is expressed by pancreatic acinar cells[40]; however, in our *Vil1-cre-tdTomato* mice we found no VIL1-expressing cells in the pancreas (Fig. 5a). In addition, histological analysis of *Vil1-cre-Gp2^{flox/flox}* ($Gp2^{IEC}$) mice revealed that they possessed both pancreatic and luminal GP2 (Fig. 5b). These results clearly indicated that $Gp2^{IEC}$ mice lacked GP2 expression by M cells, meaning that their luminal GP2 was derived only from the pancreas. Importantly, when colitis was induced in the $Gp2^{IEC}$ mice, we found no differences in body weight loss, colon length, or epithelial damage compared with control mice (Fig. 5c–e).

Next, we conducted an experiment using $Gp2^{Panc}$ mice and confirmed that PTF1A is not expressed in the intestinal compartment of *Ptf1a^{Cre-ERTM}-tdTomato* mice by using immunofluorescence microscopy (Fig. 6a). In tamoxifen-treated $Gp2^{Panc}$ mice, GP2 expression was clearly diminished in the pancreas and the luminal side of the digestive tract (Supplementary Fig. 13). $GP2^{Panc}$ mice administered DSS showed significant body weight loss as well as colon shortening with marked epithelial damage and neutrophil infiltration (Fig. 6b–e), which were also observed in $Gp2^{-/-}$ mice (Fig. 4). In addition, the population of adhesive or invasive bacteria was markedly elevated in $GP2^{Panc}$ mice with colitis compared with in control mice (Fig. 6f). These results show that GP2 derived from pancreatic acinar cells is the major source of intestinal GP2, and that GP2 expressed by M cells is not involved in the response to colitis-associated intestinal inflammation.

Together, these results suggest that the main source of GP2 is the pancreas, not M cells, and that GP2 limits the severity of colitis-associated intestinal inflammation.

**Interaction between pancreatic GP2 and intestinal *E. coli*.** Previous reports have shown that human and mouse GP2 both bind to FimH, a component of type I fimbria expressed by gram-negative bacteria[18,41]. FimH is required for bacterial binding to

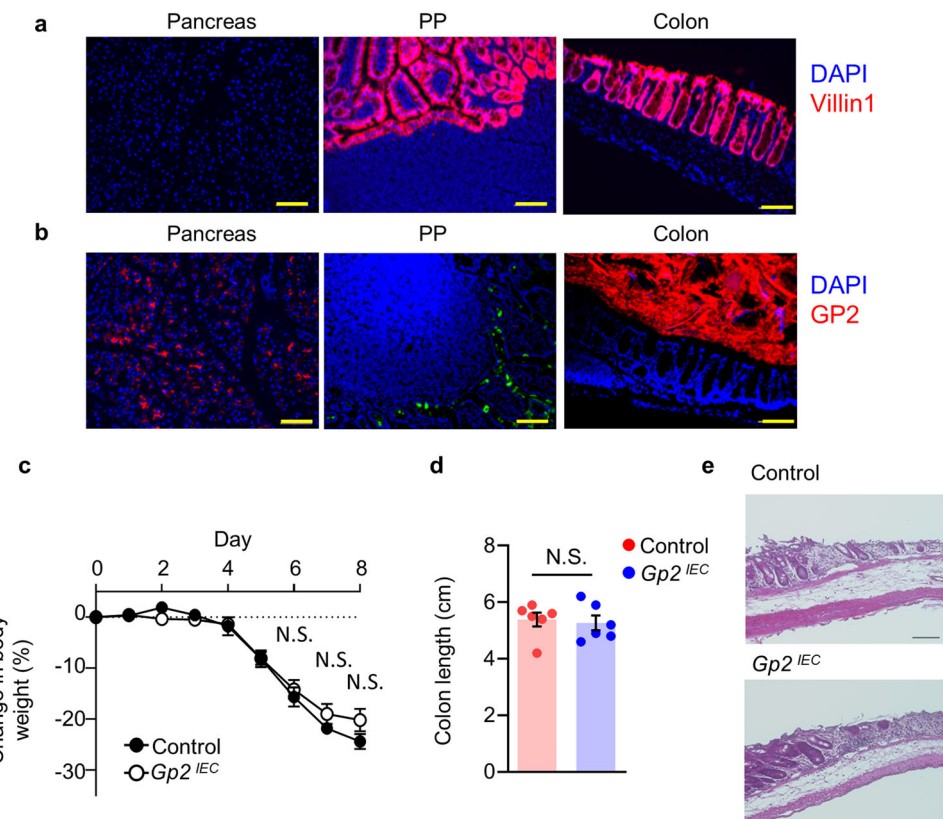

**Fig. 5 DSS-induced acute colitis in mice lacking GP2 in gut-associated lymphoid tissue. a** VIL1 expression in pancreas, Peyer's patch (PP), and colon of *Vil1-cre-tdTomato* mice were shown. **b** GP2 (red) distribution in *Vil1-cre-Gp2flox/flox* (*GP2IEC*) mice were shown. Green color indicates UEA-1 in PP. Representative data of three independent experiments are shown. Scale bars, 100 μm. **c** Body weight change after induction of acute colitis in *GP2IEC* and control (WT *Gp2flox/flox*) mice (*n* = 7 /group). N.S. indicates not significant (two-tailed unpaired *t*-test). Data are presented as mean values ± SEM. **d** Colon length in *GP2IEC* and control mice were shown (*n* = 7 /group). N.S. indicates not significant (two-tailed unpaired *t*-test). Data are presented as mean values ± SEM. **e** Hematoxylin and eosin staining of colon tissues were shown. Scale bars, 100 μm. Data are representative of three independent experiments. Source data are provided as a Source Data file.

epithelial cells, and FimH antagonists (e.g., mannosides) selectively eliminate *E. coli* from the gut[52]. Therefore, pancreatic GP2 may also have the ability to bind fecal bacteria and limit bacterial invasion.

First, we confirmed co-localization of luminal GP2 and luminal bacteria (EUB338+) in the intact colon (Fig. 7a). A similar co-localization was also observed in the ilium of the small intestine (Supplementary Fig. 14). We next examined and quantified the amounts of free and bacteria-bound luminal GP2 by means of an enzyme-linked immunosorbent assay (ELISA). Fecal samples were dissolved in phosphate-buffered saline (PBS) and then divided into fecal bacteria-bound and unbound supernatants by centrifugation. When the concentrations of unbound GP2 were measured and compared between WT mice with and without colitis, the levels of unbound GP2 were comparable between the two groups (Fig. 7b).

To further examine whether pancreatic GP2 binds to luminal bacteria, flow cytometry analysis was performed. Fecal contents from *Gp2−/−* mice were incubated with recombinant GP2 (rGP2) and stained with anti-GP2 antibody. Flow cytometric analysis revealed that in *Gp2−/−* mice about 25% of the fecal bacteria were bound to GP2, and this was further increased to about 36% in colitis mice (Fig. 7c, d). Similarly, in WT mice the percentages were 16% and 44% for intact and colitis mice, respectively. These results indicated that the GP2-associated bacterial population is increased during colitis, implying either that there is an increase in luminal GP2 concentration or an increase in GP2-associated

bacteria. To examine the size of the *E. coli* population, we sorted the GP2-bound bacteria by means of magnetic-activated cell sorting (MACS) (Fig. 7e). In intact mice, without the addition of rGP2 prior to MACS, *E. coli* was undetectable. However, in WT mice with colitis, even without the addition of GP2, *E. coli* were detected, indicating that the GP2-bound *E. coli* population was increased during colitis.

Together, these results indicate that GP2 originating from pancreatic acinar cells binds to *E. coli* bacteria in the steady state (Fig. 7d). The development of colitis results in an increased GP2-bound bacteria ratio compared with the steady state, contributing to worsening of inflammation (Fig. 7c, d).

The addition of rGP2 further increased the number of GP2-bound fecal bacteria (Fig. 7d), suggesting that not all commensal bacteria are bound to GP2 despite the elevated production of pancreatic GP2 during intestinal inflammation; this raises the possibility that rGP2 could be used to modulate the intestinal environment for the control of inflammation.

**Recombinant GP2 prevents *E. coli* invasion in mice with intestinal inflammation.** To confirm our hypothesis that pancreatic GP2 prevents bacterial infiltration, GFP-expressing *E. coli* strain S-17 (S-17-GFP) were pre-incubated with mouse rGP2 and then their binding was confirmed by staining with anti-GP2 antibody using flow cytometry analysis. This analysis confirmed that rGP2 bound to S-17-GFP (Fig. 8a). When *E. coli* S-17-GFP pre-treated or untreated with rGP2 were cultured, no significant

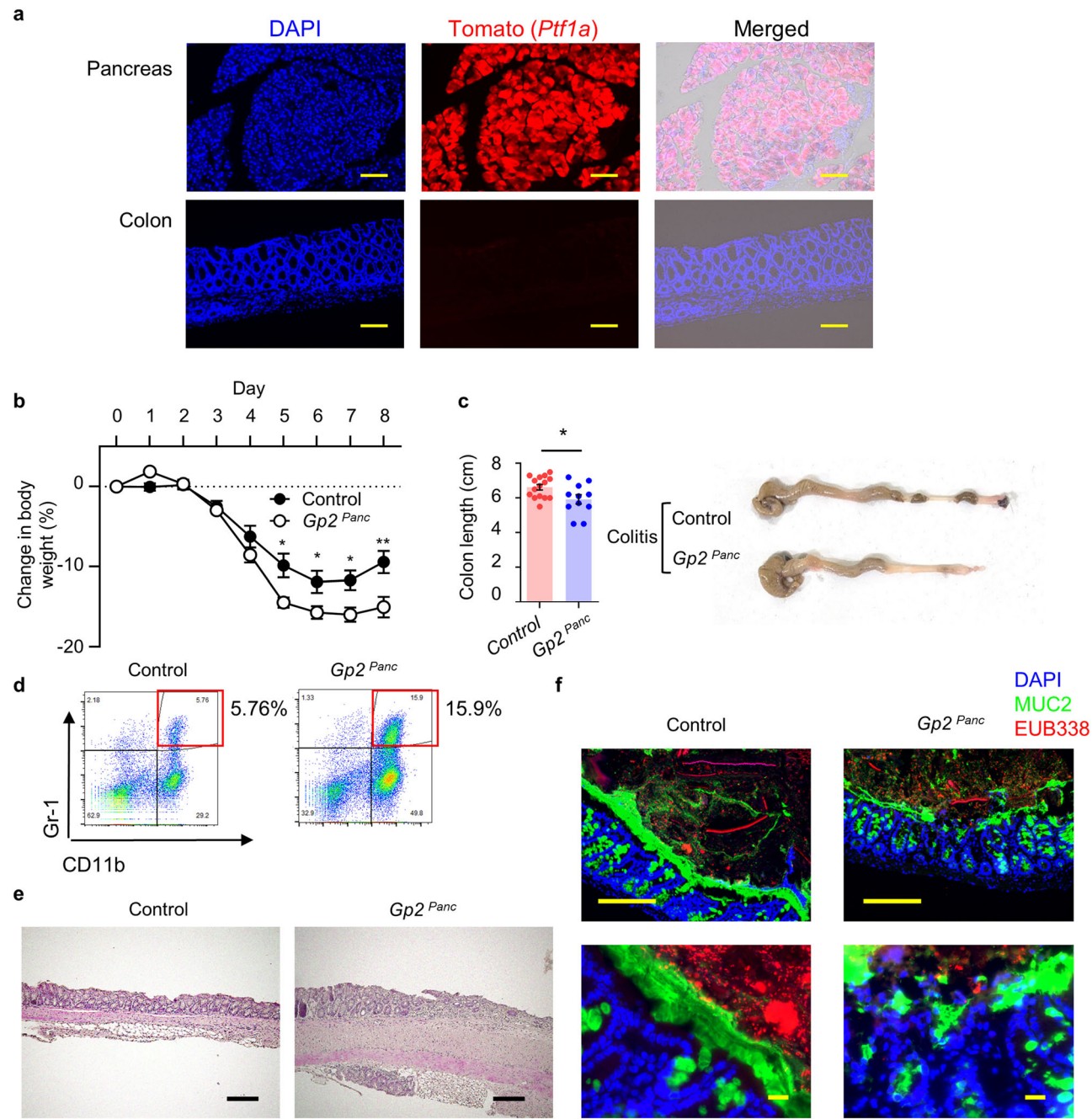

**Fig. 6 Pancreatic GP2 is important for mucosal protection. a** PTF1A-expressing cells (red: tomato) in the pancreas and colon of *Ptf1a-cre^ETRM-tdTomato* mice. Scale bars, 100 µm. Data are representative of three independent experiments. **b** Body weight changes after the induction of acute colitis in *Ptf1a-cre^ETR Δ/+-Gp2^flox/flox* (*Gp2^Panc*) (*n* = 15) and control (wild-type; *Ptf1a-cre^ETR +/+-Gp2^flox/flox*) (*n* = 11) mice, receiving tamoxifen **\*\***p < 0.01, *\*p < 0.05 (two-tailed unpaired *t*-test). Data are presented as mean values ± SEM. **c** Changes in colon length and photographs of representative colons of *Gp2^Panc* (*n* = 15) and control (*n* = 11) mice. Scale bars, 100 µm. *\*p = 0.026 (two-tailed unpaired *t*-test). Data are presented as mean values ± SEM. **d** Flow cytometric analysis of neutrophils that had infiltrated the colon in *Gp2^Panc* and control mice; representative data are shown. **e** Hematoxylin and eosin staining of colon at day 8 of DSS treatment in *Gp2^Panc* and control mice. Scale bars: 100 µm. Data are representative of three independent experiments. **f** Immunohistochemical analysis of colitis colon and luminal contents in *Gp2^Panc* and control mice. Bacteria; EUB338 (red), MUC2 (green), DAPI (blue). Scale bars: upper panel, 100 µm; lower panel, 20 µm. Data are representative of three independent experiments. Source data are provided as a Source Data file.

alteration of bacterial growth was detected (Fig. 8b), demonstrating that the binding of rGP2 did not influence bacterial growth.

In addition, we conducted experiments with FimH-deficient *E. coli* to confirm whether human rGP2 also binds specifically to FimH. When FimH-positive or -deficient *E. coli* were exposed to human rGP2, the human rGP2 bound to FimH-positive, but not FimH-deficient, *E. coli* (Fig. 8c).

To directly demonstrate that the binding of rGP2 to *E. coli* S-17-GFP leads to the blockade of bacterial attachment and invasion in colitis, a colon loop assay using DSS-treated *Gp2^−/−* mice was conducted. Colon loops from *Gp2^−/−* mice with colitis received *E. coli* S-17-GFP showing mucosal invasion of bacteria with green fluorescent signals (Fig. 8d). If *E. coli* S-17-GFP were treated with rGP2 prior to the loop assay, histological

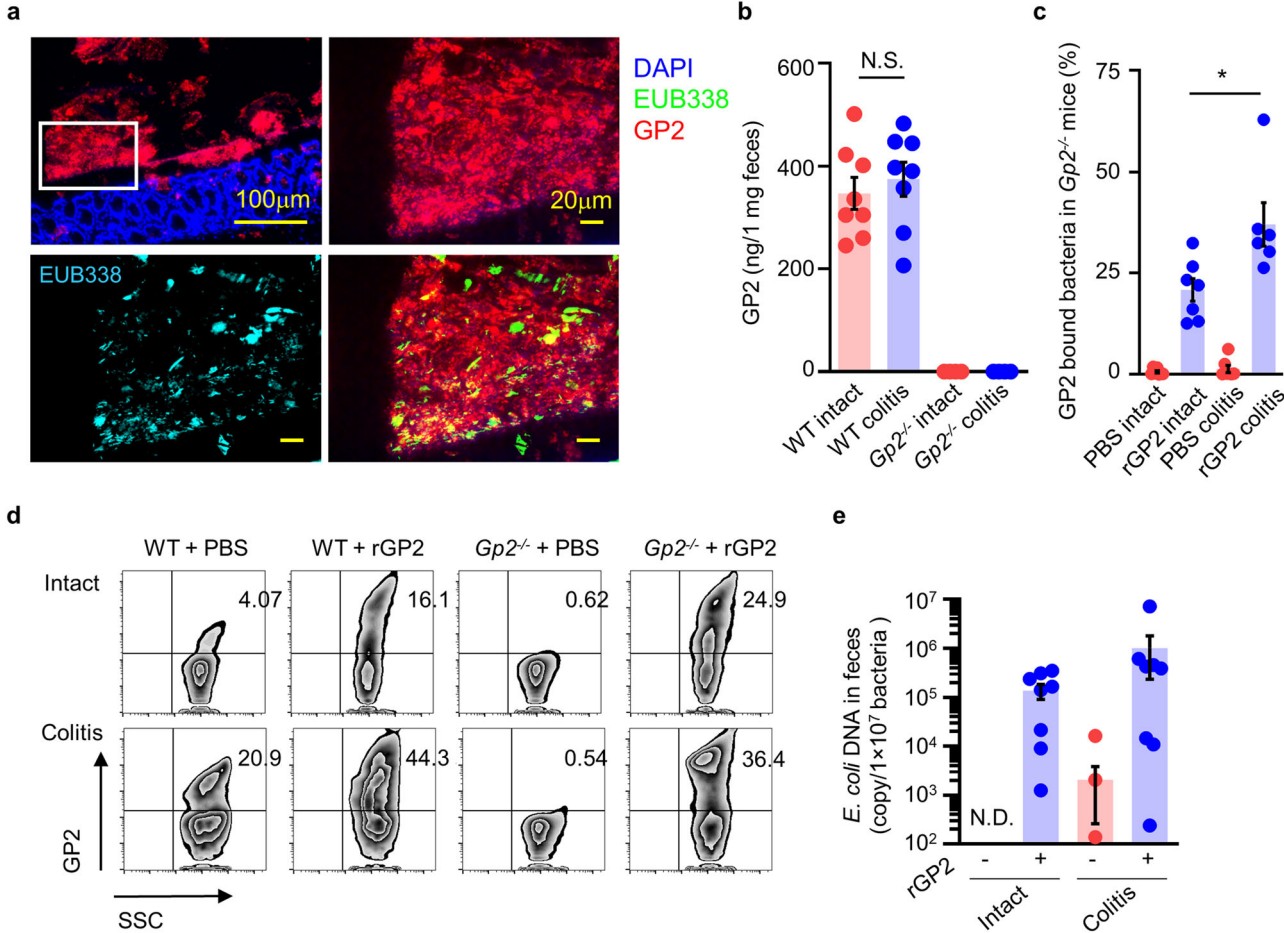

**Fig. 7 Recognition of intestinal bacteria by pancreatic GP2. a** Luminal contents of colon were stained with GP2 (red) and EUB338 (blue in left bottom and green in right bottom) were shown. Data are representative of three independent experiments. **b** Fecal unbound GP2 in WT and $Gp2^{-/-}$ mice with or without acute colitis were measured by ELISA (WT, $n = 8$; $Gp2^{-/-}$, $n = 4$). N.S. indicates not significant (two-tailed unpaired $t$-test). Data are presented as mean values ± SEM. **c** Percentage of rGP2-bound fecal bacteria isolated from $Gp2^{-/-}$ mice, PBS and rGP2 Intact $n = 7$, PBS colitis ($n = 7$), rGP2 colitis ($n = 6$). *$p = 0.017$ (two-tailed unpaired $t$-test). Data are presented as mean values ± SEM. **d** Representative flow cytometry analysis of fecal bacteria isolated from WT and $Gp2^{-/-}$ mice with or without colitis were shown. **e** *E. coli* number of non-selected (rGP2 MACS −) and selected by MACS (rGP2 MACS +) fecal bacteria ($n = 9$ /group). N.D.: not detected. Data are presented as mean values ± SEM. Source data are provided as a Source Data file.

analysis and CFU assay revealed that the bacterial invasion was significantly reduced when compared with mice received *E. coli* S-17-GFP without rGP2 (Fig. 8e, f).

Anti-GP2 autoantibodies are found in patients with inflammatory bowel disease[4–6]. It is therefore possible that translocation of GP2-bound bacteria to the systemic compartment triggers anti-GP2 autoantibody production in the intestinal lumen, which then exacerbates inflammation by neutralizing luminal GP2 function. To ensure the generation of anti-GP2 autoantibody production in vivo, we systemically administered recombinant GP2 together with complete Freund's adjuvant to $Gp2^{-/-}$ mice, and upregulation of luminal anti-GP2 IgG production was confirmed (Fig. 8g). We next examined the concentration of luminal anti-GP2 auto-IgG antibodies in the stool of GP2-immunized $Gp2^{-/-}$ mice and found elevated production of luminal anti-GP2 IgG from day 27 after immunization (Fig. 8g). When DSS-treated WT mice were examined, a significant increase of luminal anti-GP2 IgG was also noted (Fig. 8g), which is consistent with data reported for human inflammatory bowel disease[4–6]. Together with the observed increases of both anti-GP2 autoantibodies and anti-*E. coli* antibodies in DSS-treated mice (Fig. 4i), these findings suggest that translocation of GP2-bound bacteria to the systemic compartment induces anti-GP2 antibody production.

To further examine this possibility, we systemically introduced GP2-bound *E. coli* to naive mice as a model of the infiltration of GP2-bound bacteria from the gut to the systemic compartment for the production of anti-GP2 autoantibodies (Fig. 4i). As in the $Gp2^{-/-}$ mouse study (Fig. 8g), the level of anti-GP2 autoantibodies was elevated in WT mice systemically exposed to GP2-bound *E. coli* (Fig. 4i). Taken together, these results provide the first experimental evidence that infiltrating GP2-bound bacteria induce autoantibodies against GP2 under the inflammatory milieu; therefore, it is possible that anti-GP2 autoantibodies in the luminal compartment regulate GP2 function.

Finally, we conducted a rGP2-binding blocking assay using *E. coli* strain S-17 and luminal contents containing anti-GP2 IgG (Fig. 8h). Pretreatment of rGP2 with the luminal contents reduced the binding between GP2 and S-17. Together, these results indicate that once GP2-mediated inhibition of bacterial translocation is disrupted, an inflammatory loop-mediated by anti-GP2 autoantibodies is induced that worsens gut inflammation.

In summary, we found that pancreatic GP2 contributed to the prevention of bacterial adhesion to the intestinal mucosa, indicating that GP2 acts as a first line of defense in controlling the severity of colitis (Supplementary Fig. 16). Additional findings from our mouse and human studies indicate that the present data

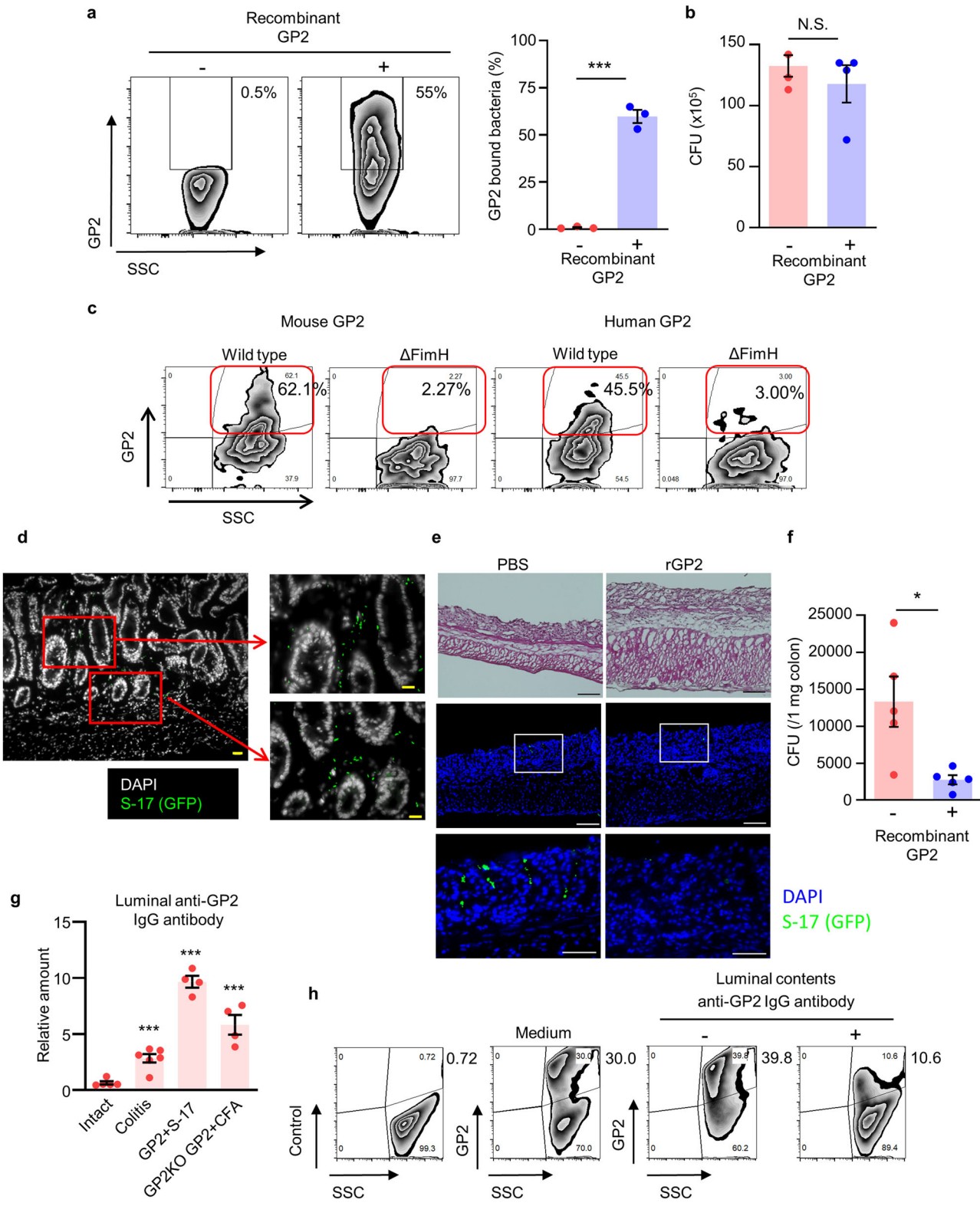

are translatable to the GP2-mediated control of human intestinal inflammation.

## Discussion

GP2 is secreted by the pancreas and is expressed on M cells located in the FAE of Peyer's patches in the small intestine[6,7]. GP2 binds to FimH and is therefore reported to be a FimH receptor and a key luminal antigen sampling molecule for the initiation of Peyer's patch-mediated antigen-specific mucosal immunity[43]. The pancreas is a major source of intestinal GP2[15]. Deficiency of GP2 results in no morphological alteration or defects in secretion of digestive enzymes (e.g., amylase) from the pancreas[46]. Therefore, the exact role of pancreatic GP2 remains unclear. We hypothesized that luminal GP2 originating from the pancreas contributes to the control of intestinal bacteria, because the GP2 homolog

**Fig. 8 Pancreatic GP2 regulates *E. coli* invasion in colitis. a** Flow cytometry of *E. coli* strain S-17-GFP bound with or without rGP2 were performed. Representative data of three independent experiments were shown. ***$p < 0.0001$ (two-tailed unpaired $t$-test). Data are presented as mean values ± SEM. **b** CFU assay of S-17-GFP incubated with recombinant GP2 ($n = 4$). N.S. indicates not significant (two-tailed unpaired $t$-test). **c** Approximately $1 \times 10^6$ CFU of WT S-17 cells and FimH-deficient S-17 cells (ΔFimH) were incubated with 1 µg of mouse or human GP2, and FACS analysis was conducted with anti-GP2 antibody. Representative data of three independent experiments are shown. **d**, **e** Immunohistochemical analysis of colon specimens isolated from ligated colon of S-17-GFP, pre-incubated with PBS or recombinant GP2 were shown. Scale bars, 50 µm (left panel) and 10 µm (right panels). **f** CFU assay of S-17-GFP injected looped colon (2% DSS day 6), $n = 5$ /group were shown. *$p = 0.017$ (two-tailed unpaired $t$-test). Data are presented as mean values ± SEM. **g** The amount of anti-GP2 IgG in the luminal contents of intact mice ($n = 5$), DSS-treated mice ($n = 6$), mice systemically administered GP2-bound heat-killed (70 °C, 30 min) S-17 cells ($n = 4$), and *GP2* knockout (GP2KO) mice immunized with rmGP2 and complete Freund's adjuvant (CFA) as positive controls was determined by ELISA on day 27 after immunization ($n = 4$). ***$p < 0.01$ (one-way ANOVA). Data are presented as mean values ± SEM. **h** Neutralizing assay for the binding of rGP2 to S-17 cells. Medium (Tris-HCL). Luminal contents from anti-GP2 IgG antibody–negative (intact) or -positive (DSS-treated) mice were pre-incubated with GP2 and added to S-17 cells. The GP2-bound SYTO9$^+$ S-17 population is shown. Data are representative of two independent experiments. Source data are provided as a Source Data file.

Tamm–Horsfall protein recognizes uropathogenic *E. coli* and prevents the binding of *E. coli* to urothelial cells[23]. Here, we show that intestinal GP2 expressed by pancreatic acinar cells controls bacterial invasion into intestinal epithelial cells, especially those in an inflammatory environment.

Luminal GP2 was observed in WT and $Gp2^{IEC}$ mice, which expressed pancreatic but not M cell GP2 (Figs. 1, 2, and 5). VIL1 is an intestinal epithelium marker[51] and we confirmed that VIL1 was specifically expressed by epithelial cells in the small and large intestines of those mice (Fig. 5a). Pancreatic acini are also reported to be VIL1-expressing cells[40]; however, in the present study no VIL1 signal was observed (Fig. 5a). Other secretory glands, including the sublingual glands, submandibular glands, conjunctiva, and tear ducts, are also reported to express or secrete GP2[27]; however, we did not observe GP2 in the salivary glands (Supplementary Fig. 2), implying that the primary source of luminal GP2 is pancreatic acinar cells.

GP2 production by acinar cells was upregulated by the inflammatory signaling of TNF (Figs. 2 and 3). GP2 expressed by acinar cells goes through membrane release and granular condensation and is then secreted into the pancreatic juice[53]. In the present study, we observed that a high amount of GP2 is secreted into the digestive tract and feces in the steady state (Figs. 1 and 2). Therefore, we predicted that together with the increase of GP2 expression, GP2-secretory pathways would also be altered in colitis. However, the expression levels of genes encoding enzymes that contribute to the cleavage of the GPI anchor of GP2 from the granule membrane (e.g., phospholipases, carboxypeptidase, and trypsin) were unaltered in mice with colitis (Supplementary Fig. 6b)[36,37]. We also found no alteration of the expression of genes encoding proteins such as small GTPases (e.g., Rab1a)[32] and trafficking proteins (syntaxins, dynamins, and myosins) that are related to the granular transport system (Supplementary Fig. 6a and data not shown)[33–35]. Past studies imply that pancreatic GP2 is constitutively released regardless of digestive signaling[46]. Therefore, it is possible that the increased production of GP2 by acinar cells with constitutive granular release leads to increased concentrations of luminal GP2 after inflammatory cytokine stimulation (e.g., by TNF) (Figs. 2c–e and 3).

Indeed, gastrointestinal diseases are often caused by the ingestion of food contaminated with pathogenic bacteria[54]. Therefore, it is possible that constitutive coverage of the digestive tract with pancreatic GP2 prevents bacterial penetration into the intestinal mucosa. If this is true, the protection of the intestinal mucosa would be an additional role played by an accessory digestive organ (i.e., pancreas).

Our 16S rRNA analysis data indicated that neither GP2 full deficiency nor conditional deficiency affected the overall profile of the luminal commensal microbiota (Fig. 4b and Supplementary Fig. 11). However, we noted that GP2 deficiency resulted in increases of select bacteria such as *E. coli* in the mucosal compartments (Fig. 4c). A MACS analysis of GP2-bound luminal bacteria (Fig. 7e) cultured under aerobic conditions revealed that *Lactobacillus spp.* (e.g., *L. johnsonii*, *L. intestinalis*, and *L. murinus*) and *Faecalibaculum rodentium* are GP2-bound bacteria also (Supplementary Fig. 15). It has been reported that systemic immune responses to the cell wall components (e.g., polysaccharides) of *L. johnsonii* are upregulated in inflammatory bowel disease patients, suggesting that translocation of *L. johnsonii* occurs via disruption of the mucosal barrier[55]. *F. rodentium* is a mucus bacteria found inside colon tumor[56]. We did not find any differences between the populations of these unknown GP2-bound bacteria in the luminal compartment between WT and GP2-deficient mice (Supplementary Fig. 10); however, it is possible that translocation of these GP2-bound bacterial species may lead to the generation of GP2 autoantibodies in colitis. Furthermore, although pancreatic GP2 has little effect on the diversity of the luminal microbiota, it plays a crucial role in the control of the interaction between the host and commensal (or pathogenic) bacteria.

We found that *Gp2* expression was significantly increased in the pancreas of mice with colitis (Fig. 2c, d), indicating that an inflammatory signal (i.e., TNF) from the intestinal mucosa stimulates the pancreas to increase the expression of GP2 (Fig. 3d,e) and subsequently the amount of GP2 in the secretory granules (Fig. 3f) to prevent excess inflammation caused by bacterial stimulation. These observations indicate the existence of a pancreatic–colon axis in the inflammatory condition. In this context, *E. coli* bacteremia is reported to pathologically augment the production of TNF, an inflammatory cytokine that plays a key role in orchestrating the inflammatory cascade in CD[57]. Patients with CD have increased accumulation of mesenteric white adipose tissue[58,59], and mesenteric fat produces a high amount of pro-inflammatory cytokines (e.g., TNF and IL-6) under inflammatory conditions[58]. Here, we found a significant increase of luminal GP2 in patients with CD (Fig. 2f). We previously reported that mesenteric fat from DSS-treated mice is a major source of TNF that leads to epithelial leakage[60]. In addition, it is possible that TNF production is systemically elevated under the inflammatory condition (Fig. 3g) and that circulating TNF accumulates in the pancreatic vasculature, where it directly stimulates acinar cells for the production of GP2 (Fig. 3g–i).

It was previously reported that GP2 expressed on Peyer's patch M cells does not bind to FimH-deficient *E. coli*[18]. FimH adhesin of type I pili is important for the adherence of *E. coli* to the intestinal epithelium, leading to colonization[52]. Epithelial binding by type I pili recognizing terminal mannoses on epithelial glycoproteins is inhibited by mannosides[52,61]. Because type I pili regulate bacterial adhesion to the epithelium and are associated with aggravation of gut inflammation[52], FimH is considered a

novel therapeutic target for the treatment of CD[62]. Indeed, FimH antagonists (e.g., monovalent heptyl mannose derivatives, monovalent thiazolyl amino-mannosides, and *n*-heptyl α-D-mannose-based glycopolymers) have been shown to be effective anti-adhesive agents against adherent-invasive *E. coli*[62]. These findings support our hypothesis that pancreatic GP2 acts as a homeostatic self-protective agent preventing bacterial adhesion to intestinal epithelial cells. Elevating luminal GP2 could be beneficial; on the other hand, the very patients one may want to deliver GP2 as a therapeutic to may already have neutralizing autoantibodies. Also, if there is pre-existing immunity against GP2, administration of additional GP2, even if luminary, may act as a "booster" and actually be detrimental. Therefore, it is forethoughtfully examined the GP2 therapy to the GP2 autoantibodies-negative patients.

In the present study, we found that pancreatic GP2 is a key molecule that interferes with the adherence of *E. coli* FimH to colon epithelial cells (Fig. 8).

Similarly, colonic epithelium-derived Ly6/PLAUR domain-containing 8 (Lypd8) protein prevents flagellated bacterial from invading the colonic mucosa[63]. Lypd8, which is derived from epithelial cells located in the upper part of the colon, segregates flagellated bacteria, including *E. coli* and *Proteus mirabilis*, from the colonic epithelium. A lack of Lypd8 results in excess intestinal inflammation after DSS treatment[63]. Together, these findings indicate the presence of both top-down protection from the pancreas to the digestive tract mediated by pancreatic GP2 and bottom-up protection by epithelial cell-derived Lypd8 for the prevention of bacterial invasion into the intestinal and colonic mucosae.

We found that the ratio of GP2-bound bacteria was significantly increased in colitis, even when abundant amounts of GP2 were secreted by the pancreas (Fig. 7c, d). In addition, we found that the number of GP2-bound bacteria was further increased by incubation with additional rGP2, indicating that the fecal bacterial population was not saturated with GP2, even though the production of pancreatic GP2 was increased during colitis (Figs. 2d, e and 7c, d). This suggests that it might be possible to use GP2 as a novel therapeutic molecule for the control of intestinal inflammation. For example, oral administration of GP2 with a suitable delivery system may be an attractive strategy to prevent the tissue inflammatory damage caused by FimH-positive bacteria in colitis. In addition, the delivery of a novel target molecule for the induction of GP2 secretion by the pancreas may be an alternative tactic for the control of gut inflammation. It may also be possible to consider combination treatment with FimH antagonists such as monovalent heptyl mannose derivatives, because these molecules possess antagonistic activity[62]. Our present study using a mouse experimental system provides the scientific foundation for the development of a pancreas–gut barrier axis-based GP2-mediated defense system for the control of the penetration of commensal and/or pathogenic bacteria through the intestinal epithelium during intestinal inflammation, which will useful for developing potential preventive or therapeutic therapies for human intestinal inflammation. The possibility of translating our findings to the clinic is further supported by our experiment using human rGP; when we examined the reactivity of human rGP2 against FimH-positive and -negative *E. coli*, we found that human rGP2 specifically bound to FimH (Fig. 8c).

As demonstrated in the present study, GP2 is a key mucosal defensive molecule, especially under the pathological condition of intestinal inflammation when the mucus layer is reduced (Figs. 4 and 6), that binds to commensal bacteria and prevents their adhesion and invasion into the intestinal epithelium. Since cases of septic shock are rare in patients with inflammatory bowel disease despite their disrupted intestinal epithelial barrier, pancreatic GP2 secretion in the luminal tract might explain the low incidence of sepsis in this patient population.

Pancreatic autoantibodies, including anti-GP2 antibodies, are reported to be potentially useful as biomarkers of CD[5]. *GP2* has been shown to be an autoimmune regulator-dependent gene in thymic epithelial cells[64], indicating that the immune response to GP2 is usually immunologically tolerant and educated to suppress autoimmune responses. Therefore, the loss of tolerance to GP2 might be associated with the initiation and triggering of auto-immunity. Although clinical evidence suggests that GP2 auto-antibodies play a critical role in human intestinal inflammation, and therefore that neutralizing or blocking the GP2-mediated gut defense system may have clinical applications[65], the mechanisms of GP2 autoantibody production in CD are yet to be fully elucidated. Our present results indicate that systemic antibodies to commensal bacteria (e.g., *E. coli*) are induced during colitis (Fig. 4i) and that translocated GP2-conjugated bacteria might act as immunopotentiators for the induction of an autoimmune response to GP2. Indeed, administration of GP2-bound heat-killed (70 °C, 30 min) *E. coli* resulted in the upregulation of luminal anti-GP2 autoantibody production (Fig. 8g). Importantly, as reported in human patients[6], intestinal inflammation was found to lead to anti-GP2 autoantibody production in mice (Fig. 8g).

It has been suggested that FimH might be a promising target to prevent bacterial penetration, including that by adherent-invasive *E. coli*, and the associated intestinal inflammation[66]. However, anti-GP2 autoantibodies may disrupt the defensive effects of GP2 by blocking the pancreatic GP2-mediated defense against adherent-invasive *E. coli*, which could increase the severity of the disease. Even though we did not purify antibodies to GP2 from the luminal contents of mice with colitis, our findings revealed that pre-incubation with luminal contents containing anti-GP2 autoantibodies resulted in a reduction of GP2 binding to *E. coli* (Fig. 8h).

Although acute or chronic pancreatitis are rare complications in CD and are mainly caused by the development of gallstones or by therapeutic drugs given to ameliorate the symptoms of the disease, pancreatic involvement prior to the onset of CD has also reported[8,67]. Such pancreatic involvement might be associated with GP2 dysfunction or increased autoantibodies and may induce the development of CD. Indeed, serum GP2 levels are increased in pancreatitis animal models and patients with CD[68,69]. Further studies are needed to reveal the roles of GP2 and anti-GP2 autoantibodies in the context of the relationship between pancreatitis and CD.

In summary, the present results show that GP2 secreted by the pancreas into the intestinal lumen recognizes FimH expressed on *E. coli*. A lack of pancreatic GP2 resulted in greater bacterial invasion in the large intestine in mice with colitis compared with in intact mice. Administration of rGP2 prevented *E. coli* from invading the intestinal epithelium in mice with colitis. Thus, these results show the critical role of pancreas and intestinal barrier axis in which pancreatic GP2 acts as part of the first line of defense against intestinal bacterial invasion during gut inflammation.

## Methods

**Mice**. Eight- to 10-week-old male C57BL/6 mice were used in this study. All mice in this study were on a C57BL/6 background mice. *Vil1-cre* (#004586), *Ptf1a-cre^ERTM* (#019378), *Rosa26-tdTomato* (#007914), mice were purchased from Jackson Laboratory. *Col1a2-GFP* mice were provided by Y.I.[70] *Gp2^−/−* mice have been generated as described previously[6]. *Gp2^flox/flox* mice were constructed by K.H. and H.O. and construction is mentioned below (Supplementary Fig. 17). Acute colitis was induced in littermate or age-matched mice, co-housed from 2 weeks of age, by adding 2.0–2.25% DSS (MP Biomedicals #160110) to the drinking water for 5 days; the mice were euthanized at day 6 or day 8. The loss of 30% of initial body

weight was considered to be fatal, and such mice were euthanized. Mice were treated with 250 mg/kg body weight (i.p.) of tamoxifen (Sigma-Aldrich #T5648) for 5 consecutive days prior to DSS treatment and continue for every 2 days. To examine the roles of GP2 in early life, 2-week-old mice were treated with 125 mg/kg body weight (i.p.) of tamoxifen for 5 consecutive days (Supplementary Fig. 11a). To deplete commensal bacteria, mice received broad-spectrum antibiotics [ampicillin (1 g/L), vancomycin(0.5 g/L), neomycin sulfate(1 g/L), metronidazole(1 g/L)] in the drinking water for 4 weeks, as previously described[71].

TNBS-induced colitis mice were generated as described previously[50]. Briefly, the skin on the back of mice was sensitized with a 2.5% mixture of TNBS (Sigma-Aldrich) in acetone and olive oil. At 1 week after sensitization, after a 4-h fast, the mice were intra-rectally administered 100 µL of 2.0% TNBS in 50% ethanol. To induce anti-GP2 antibodies 1 mg/mL of recombinant mouse GP2 mixed with complete Freund's adjuvant was subcutaneously immunized to GP2-deficient mice. Stools were obtained at 4 weeks after immunizations.

All mice except germ-free mice were maintained under specific-pathogen-free conditions at the experimental animal facility of the Institute of Medical Science, The University of Tokyo and Chiba University, Japan. All experiments were approved by the Animal Care and Use Committee of the University of Tokyo and Chiba University.

**Generation of _Gp2_ mutant mice**. The target vector containing a 2.3-kb genomic fragment (intron1 to intron2) floxed in intron1, neomycin resistant gene floxed at both ends, a 3-kb genomic fragment (intron2 to exon3) and the HSV-tk gene, was constructed with pBluescript II SK (+), as depicted in Supplementary Fig. 17. The linearized targeting vector was introduced into embryonic stem cells to obtain the clone with homologous recombination. The floxed _Gp2_ mice were backcrossed onto C57BL/6J genetic background at least for eight generations, and then were crossed with transgenic mice expressing _Vil1-Cre_ or _Ptf1a-Cre_ERTM to generate conditional Gp2-deficient mice.

**Immunohistochemical and histological analysis**. Pancreas, Peyer's patches, large intestine, gastrointestinal ducts (stomach, small intestine, and large intestine) including luminal contents, submandibular glands and sublingual glands were fixed in 4% paraformaldehyde (Wako) and embedded in paraffin or followed by 15% sucrose cryoprotection at 4 °C overnight and embedded in OCT compound (Sakura Finetek Japan). Large intestine without luminal content was fixed in 4% paraformaldehyde and embedded in paraffin. After embedding, 5-µm sections were cut and stained with the following antibodies: for GP2 detection, anti-mGP2 (MBL, 2F11-C3, #D278-3, 1:200) and Alexa 555-conjugated anti-rat IgG (BioLegend, Poly4054, #405420, 1:200). To confirm that GP2 signal was correct, serial sections were stained with isotype control (rat IgG2a, BioLegend, RTK2758, #400501, 1:100) and Alexa 555-conjugated anti-rat IgG in all samples; for Peyer's patch FAE, fluorescein isothiocyanate (FITC)-conjugated UEA-1 (Vector Laboratories, # FL-1061, 1:100). DAPI (4′,6-diamidino-2-phenylindole; Dojindo, #D523, 1:1000) was used for counter staining. Observations were performed under a fluorescence microscope (BZ-9000; Keyence). Hematoxylin (Wako) and eosin (Wako) staining was performed by using standard procedures.

**Collection of pancreatic juice and luminal wash**. The protocol was modified from that of a previously described study[72]. Mice were anesthetized with isoflurane (Muromachi Kikai #MK-AT210), and right upper quadrant laparotomy was performed. Sutures were tied tightly around the duodenum just distal to the pylorus, as described previously[72], and ~1.5 cm distal to the pylorus to form a completely occluded intestinal loop. A 2.5-cm polyethyleneSP-45 tube (Natsume Seisakusho) was placed into the midportion of this loop of duodenum through a small nick created in the bowel wall by using a 21-gauge needle. Fluid was collected into the tube for 15 min. To collect luminal washout for the measurement of luminal GP2, 3 cm of jejunum was excised and flushed with 250 µL PBS.

**Cell collection and fluorescence-activated cell sorting analysis**. A part of enzymatically dispersed colon cells by collagenase (Wako #032-22364) were collected as whole colon cells, and the other were incubated with 5 µg/mL of anti-CD16/32 antibody (Fc block; BD Biosciences, 2.4G2, #553141, 1:500) for 5 min and then incubated for 30 min at 4 °C with fluorescence-labeled antibodies specific for phycoerythrin (PE)-conjugated anti-podoplanin (Biolegend, 8.1.1, #127410, 1:1000), Alexa 647-conjugated anti-EpCAM (Biolegend, G8.8, #118211, 1:500), PE-Cy7-conjugated anti-CD90.2 (Biolegend, 5302.1, #140410, 1:500), and Pacific Blue-conjugated anti-CD45 (BioLegend, R30-F11, #103126, 1:100). Cell population definitions were following; Hematopoietic cells, CD45 (+); epithelium, CD45 (−); EpCAM (+), type 1 collagen (−); Fibroblasts, CD45 (−), EpCAM (−), type 1 collagen (+), CD90.2 (+), podoplanin (+); Neutrophils, [CD45 (+), Gr-1 (+) (Biolegend, RB6-8C5, #108412, 1:500), CD11b (+) (Biolegend, M1/70, #101262 and # 101224, 1:500)]. Lived cells were gated as 7-AAD (Biolegend, #420404, 1:500) negative fraction. Cell analysis was conducted with FACS CANTO (BD Bioscience) and ATTUNE Next (Thermo Fisher Scientific) and, cell sorting was conducted with a FACSAria III instrument (BD Biosciences) and data analysis were

performed with Flow Jo (Treestar Inc., V10)[50]. Gating strategies for the flow cytometry were shown in Supplementary Fig. 18.

**Quantitative polymerase chain reaction**. FAE was isolated from Peyer's patches as previously described[73]. Total RNA was prepared by using TRIzol (Invitrogen #15596018). Pancreatic RNA was collected with TRIzol as previously described and RNA integrity was evaluated by means of agarose gel electrophoresis[74]. Reverse transcription was performed by using Superscript VILO (Invitrogen #11755-500). A LightCycler 480 II instrument (Roche) and the Universal Probe Library (Roche) was used for qPCR. Gene expression levels were normalized to that of _Gapdh_ (glyceraldehyde-3-phosphate dehydrogenase). Primers sets were listed in Supplementary table.

**Immunohistochemistry and fluorescent in situ hybridization**. Colon and its luminal content were fixed in methanol–Carnoy's solution (methanol: chloroform: acetic acid = 6: 3: 1) for 3 h and then embedded in paraffin. To detect bacteria, 5-µm sections were dewaxed and incubated overnight at 40 °C with 5 µg of Alexa 647-conjugated EUB338 probe (Invitrogen) (5′-GCTGCCTCCCGTAGGAGT-3′) in hybridization buffer (0.9 mM NaCl, 20 mM Tris-HCl, 0.1% sodium dodecyl sulfate)[75]. After incubation, the sections were rinsed in wash buffer (0.45 mM NaCl, 20 mM Tris-HCl, 0.01% sodium dodecyl sulfate), blocked with 1% bovine serum albumin in PBS, and stained with anti-MUC2 antibody (Santa Cruz Biotechnology, sc15334, Lot#F1915, 1:50) and DyLight 488-conjugated anti-rabbit IgG (BioLegend, Poly4046, #406404, 1:200). Nuclei were counterstained with DAPI. Observation was performed under a fluorescence microscope (BZ-9000).

**Pancreatic acinar cell culture and stimulation**. Mice pancreatic acinar cell culture was performed as previously described[39]. Briefly, the pancreas was removed and minced in small pieces. Then the pancreatic segments were enzymatically digested in 10 mL HBSS (Nacalai Tesque, #17461-05) with 10 mM HEPES (Gibco, #15630-080), 200 U/mL collagenase (Wako), and 0.25 mg/mL trypsin inhibitor (Wako, #202-09221) for 30 min at 37 °C. During incubation, mechanical digestion using serological pipettes was performed every 10 min. After washing, pancreatic acini were resuspended in 7 mL of Waymouth's medium (Gibco, #11220-035) containing 2.5% fetal bovine serum (HyClone), 10 mM HEPES, 0.25 mg/mL trypsin inhibitor and 25 ng/mL recombinant human epidermal growth factor (PEPROTECH, #315-09), seeded on type I collagen coated six-well plate and cultured at 37 °C overnight.

Cultured pancreatic acini from control of _Ptf1a_Cre-ERTM_-tdTomato_ mice was supplemented with 10 µM of 4-hydroxy Tamoxifen (4OHT) (Sigma-Aldrich, #H7904).

For stimulation experiments, 200 ng/mL LPS derived from _E. coli_ (Sigma-Aldrich, #L2630), 20 ng/mL TNF (BioLegend, #575204), 20 ng/mL IL-6 (BioLegend, #575702), or 100 ng/mL RANKL (Wako, #184-01791) were added in medium representatively and cultured at 37 °C overnight. The stimulated pancreatic acini were resuspended in TRIzol and performed qPCR described above.

**Feces preparation**. Feces was collected, suspended in PBS (10 µL/mg), and then centrifuged (4 °C, 20,400 RCF, 5 min) to divide it into supernatants and pellet. Supernatants were collected and stored at −80 °C until use. Pellet was resuspended in 4% paraformaldehyde, and the mixture was incubated at 4 °C overnight. After centrifuging (4 °C, 200 RCF, 2 min), the supernatant was transferred to a new tube. After centrifuging again (4 °C, 20,400 RCF, 10 min), the supernatant was removed, and the pellet was resuspended in 10% glycerol and stored as bacterial samples at −80 °C until use.

**Enzyme-linked immunosorbent assay**. As previously described, _E. coli_ strain S-17[76] were freeze-dried by Techno Suruga Laboratory. Ninety-six-well plates were coated with the anti-mouse Ig(H + L) (1 µg/mL; Southern Biotech, #1010-01) to detect total IgA or freeze-dried S-17 (1 mg/mL) for detection of anti-_E. coli_ IgA, IgM, IgG and incubated at 4 °C overnight. The plates were washed, 1% bovine serum albumin (Nacalai Tesque) in PBS was added for blocking, and the plates were incubated for 1 h at room temperature. After the plates were washed, test samples were added, and the plates were incubated overnight at 4 °C. Then, the plates were washed and horseradish peroxidase-conjugated anti-mouse IgA, IgM, IgG (Southern Biotech, #1040-05, #1040-05, #1020-05, 1:4000) in 1% bovine serum albumin in PBS was added, respectively, and the plates were incubated for a further 1 h at room temperature. The plates were then washed, and TMB Peroxidase Substrate (Kirkegaard & Perry Laboratories, #5120-0050) was added. Stop solution was added after 2 min incubation, followed by reading at 450 nm with a spectrometer. A GP2 ELISA kit (MyBioSource, #MBS081334) was used to detect fecal GP2, in accordance with the manufacturer's instructions.

**16S rRNA gene sequencing analysis**. Fecal bacterial DNA for 16S rRNA gene sequencing was prepared as described previously[77]. In brief, bacterial DNA was isolated by means of the enzymatic lysis method using lysozyme (Sigma-Aldrich) and achromopeptidase (Wako). The DNA samples were then purified by treatment with RNase A (Wako) followed by precipitation with 20% polyethylene glycol solution (PEG6000 in 2.5 M NaCl). The DNA was pelleted by centrifugation, rinsed

with 75% ethanol, and dissolved in TE buffer. Metagenomic sequencing of the samples was performed on our behalf by MyMetagenome (www.mymetagenome. co.jp).

**Generation of recombinant GP2.** To obtain constructs for 8× His-tagged rGP2 (mouse), cDNA corresponding to the amino-terminal region of *Gp2* without the transmembrane domain were amplified by PCR from cDNA of mouse pancreas distinguished above. The following primer sets were used: (forward) 5′- AGGGA GACCCAAGCTGGCTAGCATGAAAAGGATGGTGGGGTT-3′ and (reverse) 5′-GTGATGATGGTGGTGATGATGATGTGTAGTGTGGGGAGTGCCCC-3′ as 1st PCR and (forward) 5′-AGGGAGACCCAAGCTGGCTAGCATGAAAAGGAT GGTGGGGTT-3′ and (reverse) 5′-GGATATCTGCAGAATTCTCAGTGATGATG GTGGTGATGATGATG-3′ as 2nd PCR. cDNA fragments were inserted into the *NheI* and *EcoRI* cloning sites of the pcDNA3.1(+) expression vector. Then, human embryonic kidney (HEK293T) cells were transiently transfected with the rGP2 expression vectors and cultured for 5 days. Recombinant protein secreted into the supernatant was purified by using a HisTALON Gravity Column Purification kit (TaKaRa).

To obtain constructs for chimeric mouse–human GP2 protein harboring the Fc segment of human IgG1 (mGP2-Fc), synthetic cDNA corresponding to the amino acid sequence of mouse *GP2* (accession no. AK009661 from NCBI Blast) lacking the signal peptide and the transmembrane domain was obtained from TaKaRa Bio Inc. and inserted into pFUSE-hIgG1e1-Fc2 plasmid.

Lentiviral particles were generated by transfection of the constructed vectors with packaging plasmid (pCAG-HIVgp) and the VSV-G/Rev-expressing plasmid (pCMV-VSV-G-RSV-Rev) into human embryonic kidney cells (HEK293T) by using Lipofectamine 2000 followed by concentration with Lenti-X Concentrator (Clontech Laboratories). The supernatant of the transfected HEK293T cells was purified by using rProtein A Sepharose Fast Flow affinity medium (GE Healthcare).

**Construction of a non-polar mutant of *E. coli* strain S-17 strain.** A non-polar deletion mutant was constructed by disrupting the *fimH* gene in *E. coli* S-17 by using the Red recombinase-mediated recombination system, as described previously[78]. Briefly, the plasmid pKD4 was used as a template to amplify a kanamycin-resistance marker flanked by *fimH*-specific sequences. The resulting PCR product was digested with DpnI and electroporated into *E. coli* strain S-17 carrying pKD46, and the transformants were grown on L-agar supplemented with 10 µg/mL kanamycin. The transformants were incubated at 37 °C to cure the pKD46, and kanamycin-resistant genes were eliminated by using the helper plasmid pCP20, which encodes FLP recombinase. The helper plasmid was subsequently cured by growth at 37 °C.

**Fecal bacterial flow cytometry.** Fecal bacterial samples were analyzed by spectrophotometer and $1 \times 10^7$ cells (assuming OD 600 $1.0 = 5 \times 10^8$ cell/mL) were fixed by 4% paraformaldehyde at 4 °C for 3 h. After centrifuge and discard supernatants, bacteria were resuspended and incubated with 10 µg rGP2 or PBS at 4 °C overnight. GP2-bound bacteria were detected by using a combination of anti-mGP2 antibody (MBL, 2F11-C3, #D278-3, D278-5, D278-6, 1:100) and PE-conjugated anti-rat IgG2a antibody (BioLegend, # 407508, 1:100) or PE-Streptavidin (Biolegend, #405203 or BD, # 554061) diluted 1:100. For human GP2, PE-conjugated anti-human GP2 (MBL, #D277-5, 1:100) were used. Live bacteria were stained with LIVE/DEAD BacLight, Bacterial Viability and Counting Kit (Invitrogen, #L34856) in accordance with the manufacturer's instructions. Flow cytometry analysis was performed with a FACSCanto II instrument (BD Biosciences) or ATTUNE Next (Thermo Fisher Scientific). Gating strategies for the bacterial flow cytometry were shown in Supplementary Fig. 19

**Bacterial selection by MACS.** Bacteria were incubated with 10 µg rGP2 at 4 °C overnight. To collect rGP2-bound bacteria, biotin-conjugated anti-mGP2 (MBL, #D278-6) antibody diluted 1:100 and anti-biotin microbeads (anti-biotin microbeads (Miltenyi Biotec, #130-090-485) diluted 1:20 were used. Positive selection was performed by using a MACS MS column (Miltenyi Biotec).

**Bacterial DNA collection and quantitative PCR of 16S rRNA genes.** Total fecal, MACS-selected bacteria, and mucosa-associated bacteria were collected in FastPrep tubes to which glass beads, 500 µL phenol/chloroform/isoamylalcohol, and 100 µL of 10% sodium dodecyl sulfate were added. The mixture was vortexed vigorously by using a FastPrep homogenizer (BIO-101; Thermo) at power setting 6.0 for 45 s. After centrifugation (20400 RCF, 4 °C, 5 min), 500 µL of the supernatant was collected. Isopropanol precipitation was used for DNA purification. For collection of mucosal bacteria, colon segments were taken from mice and opened longitudinally. The contents were removed by forceps, then colon segments were cut into 3-cm piece. Mucosal layer of colon was scraped by a germ-free slide glass. By pipetting, mucosal segment was collected in FastPrep tubes.

Quantitative PCR was performed by using a LightCycler 480 SYBR Green I Master Mix (Roche) and a LightCycler 480 instrument. As reported previously, the following primer sets were used[63]: 'Eubacteria', 5′-CGGTGAATACGTTCCCGG-3′

and 5′-TACGGCTACCTTGTTACGACTT-3′; '*E. coli*', 5′-GAGTAAAGTTAATA CCCTTTGCTCATTG-3′ and 5′-GAGACTCAAGCTKRCCAGTATCAG-3′; '*Lactobacillus*', 5′-TGGAAACAGRTGCTAATACCG-3′ and 5′-GTCCATTGTGG AAGATTCCC-3′; '*Bacteroides*', 5′-GAGAGGAAGGTCCCCCAC-3′ and 5′-CGCT ACTTGGCTGGTTCAG-3′; '*Helicobacter*', 5′-CTATGACGGGTATCCGGCC-3′ and 5′-TCGCCTTCGCAATGAGTATT-3′; '*Proteus*', 5′-GTTATTCGTGATGGTA TGGG-3′ and 5′-ATAAAGGTGGTTACGCCAGA-3′.

**In vitro bacterial flow cytometry.** Streptomycin-resistant green fluorescent protein (GFP)-expressing *E. coli* strain S-17 (S-17-GFP) was generated as described previously[75]. S-17-GFP were cultured in Luria-Bertani medium (Nacalai Tesque) with 50 µg/mL streptomycin (Wako) overnight, then bacteria were transferred into LB medium with 1 mM isopropyl β-D-1-thiogalactopyranoside (Wako) and incubated at 37 °C for an hour. After incubation, bacteria were resuspended in 800 µL of 4% paraformaldehyde at 4 °C for 3 h. Then bacteria were resuspended into PBS and measured OD 600 by spectrophotometer. A total of $1 \times 10^7$ cells (assuming OD 600 $1.0 = 5 \times 10^8$ cell/mL) bacteria were transferred to a new tube and resuspended and incubated with 10 µg rGP2 or PBS at 4 °C overnight. GP2-bound bacteria were detected by using a combination of anti-mGP2 antibody diluted 1:100 and PE-conjugated anti-rat IgG2a antibody diluted 1:100. Flow cytometry analysis was performed with a FACSCanto II instrument or ATTUNE Next. Gating strategies for the bacterial flow cytometry were shown in Supplementary Fig. 18.

**CFU assay.** Tenfold serial dilutions of S-17-GFP incubated with recombinant GP2 (10 µg per $1 \times 10^7$ cells) or PBS at 37 °C overnight were plated onto a Luria-Bertani agar (Nacalai Tesque) plate containing 50 µg/mL streptomycin and incubated overnight at 37 °C. After incubation, the number of colonies were counted and CFUs were calculated.

**Ligated colonic loop assay.** Approximately $1 \times 10^9$ cells S-17-GFP were incubated with 1 mg rGP2 or PBS at 37 °C overnight and then resuspended in 100 µL PBS. *Gp2*$^{-/-}$ mice in which colitis had been induced were anesthetized by using isoflurane. A distal colon loop was ligated, and the prepared S-17-GFP were injected into the loop. After 1 h, the mice were euthanized, and colonic tissues were collected. The tissues were washed with PBS and then divided into two portions to subject to immunohistochemistry and CFU assay. For CFU assay, weighed colon was homogenized in 1.5 mL PBS and tenfold serial dilutions were made. This protocol is based on a previous report describing a loop assay for examination of Peyer's patches[79].

**Collection of human fecal samples.** Patients in this study were recruited from Osaka University Hospital (Osaka, Japan). There were no self-selection and consented patients were recruited if they presented with inflamed mucosa. Fecal samples were obtained from CD patients (7), UC patients (9), and healthy volunteers (5) at Osaka University Hospital (Osaka, Japan). The experiments were approved by the human ethical committee of Osaka University Hospital and The University of Tokyo, and all tissues were sampled with written informed consent. Human GP2 ELISA was performed in accordance with the manufacturer's instructions (MyBioScience, #MBS9329271).

**Statistical analyses.** All statistical analyses were conducted by using GraphPad Prism 6 and 8 (La Jolla, CA, USA). No statistical methods were used to determine sample size. Results were compared by using an unpaired two-tailed Student's *t* test and nonparametric Mann–Whitney *U* test, or a one-way analysis of variance (ANOVA) followed by Dunnett's multiple comparison tests and Kruskal–Wallis test followed by Mann–Whitney *U* test were performed. Survival ratio was analyzed by two-tailed log-rank test. $p < 0.05$ was considered to be statistically significant. These experiments were not randomized and the investigators were not blinded to allocation during experiments and outcome assessment. Data are presented as mean values ± SEM.

**Reporting summary.** Further information on research design is available in the Nature Research Reporting Summary linked to this article.

## Data availability
Metagenomic 16S rRNA sequencing data have been deposited in the DNA Data Bank Japan (DDBJ) under the accession codes PRJDB10865 (http://trace.ddbj.nig.ac.jp/BPSearch/bioproject?acc=PRJDB10865) and PRJDB10886 (http://trace.ddbj.nig.ac.jp/BPSearch/bioproject?acc=PRJDB10886). All other data are available in the article and Supplementary Information files or from the corresponding author upon reasonable request. Source data are provided with this paper.

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

## Acknowledgements
We thank Dr. Anson W. Lowe (Stanford University, Palo Alto, CA) and for providing the Gp2⁻/⁻ mice and Dr. Kazunori Kadokura for helping generation of GP2 floxed mice. We thank Naoko Shibata, Akie Inami, Akemi Arakawa, Kanako Shimizu, Akiko Naito, Kayo Satonaga, Keiko Warren, and Hiroko Yamamoto for their technical and administrative assistance. This work was supported by grants from Japan Agency for Medical Research and Development (AMED) for H.K. (16jm0110012h0002 and 18ek0410032h0003) and Y.K. (PRIME: 20gm6010012h0004 and 20gm6210024h0001); Japan Society for the Promotion of Science (JSPS) for Grant-in Aid for Scientific Research S (18H05280 to H.K. and Y.K.) and Scientific Research B (19H03450 to Y.K.), Young Scientists A (16H06243 to Y.K.), Challenging Research or (Exploratory) [17K19550 and 19K22634 to Y.K.], Funds for the Promotion of Joint International Research (18KK0432 to Y.K.); The Ministry of Education, Culture, Sports, Science, and Technology (MEXT) for Translational Research Network Program (at the University of Tokyo) Seeds A (Y.K), B (H.K.), and C (H.K.), and LEADER (Y.K.); Foundations from Senri Life Science Foundation (Y.K.); Mochida Memorial Foundation for Medical and Pharmaceutical Research (Y.K.); The Takeda Science Foundation (Y.K.); The Uehara Memorial Foundation (Y.K.); The Sumitomo Foundation (Y.K.); The Naito Foundation (Y.K.); Kato Memorial Bio Science Foundation (Y.K.); Yakult Bio-Science Foundation (Y.K); JICA- SATREPS (H,K.): the Chiba University-UC San Diego Center for Mucosal Immunology, Allergy, and Vaccines (cMAV) and NIH P30 DK120515 [H.K.].

## Author contributions
Y.Kurashima and T.K. performed experiments, analyzed data, and wrote the manuscript; Y.Kurashima and H.K. designed the study, supervised, and wrote the manuscript; S.M., F.A., K.S., and M.M. performed experiments; K.H., H.O., and K.K. constructed and generated Gp2 floxed mice; Y.I. constructed and generated Col1a2-GFP mice; H.A. and T.S. constructed the ΔFimH bacteria and performed the analysis; A.S., Y.G., and Y.Y. constructed and purified recombinant GP2; N.S., Y.Kim, W.S., and M.H. performed the 16S rRNA gene analysis, H.I. examined human specimens.

## Competing interests
The authors declare no competing interests.
