## [Peer Review File · Nature Communications]

Reviewers' comments:

Reviewer #1 (Remarks to the Author):

In their manuscript "Pancreatic Glycoprotein 2 as the First Line of Defense for Mucosal Protection", Kurashima et al. describe that pancreas and not intestine-derived GP-2 is the source of luminal GP-2 and helps "opsonize" bacteria. The benefit of bacterial binding only becomes evident upon intestinal barrier breakdown, in a model of DSS induced colitis. Conversely, barrier-breakdown also leads to increased pancreatic expression of GP-2, suggesting a communication from gut to pancreas.

Though the concept of pancreas-derived antimicrobials is interesting, this is far from new for GP-2. The manuscript scratches only the surface of many aspects that could go into much more depth to answer the more interesting questions as to how e.g. auto-antibodies against GP-2 arise in IBDs and if they are at all relevant (are they found in the lumen? Do they really inhibit GP-2 function), what GP-2 may bind beyond FimH, if it is the only antimicrobial secreted by the exocrine pancreas, how the secretion is regulated on a granule level, what effect GP-2 has on the mucosal rather than luminal microbiome etc. Thus, the authors do not show much beyond what is known.

Specific technical concerns:

1. Quantification by ELISA of GP-2 in pancreatic juice and in lumen along the gut in WT mice under homeostatic conditions and upon DSS would be more powerful than IHC of the lumen.
2. Similarly WB against GP2 to show specificity of signal would be more reassuring- some staining in Gp2^{-/-} mice seen (Fig. 4h, 6f)
3. The TNF α dependence of GP-2 expression is not convincing: it is only based on 1.5 fold increase *in vitro*, and may not explain what is happening upon DSS (nor IBD patients without anti-TNF therapy, as TNF is more likely up); also secretion of GP-2 may be more important than expression. (see 1.). Can the authors show increased circulating/pancreatic TNF α upon DSS in their mice? Additionally, adding the serum of DSS treated versus controls onto the acinar cell cultures and looking at GP2 expression would be more convincing, to see if the TNF α levels, if raised at all, are sufficient. TNF α from creeping mesenteric fat is unlikely to reach acinar cells directly through the organ matrices, while the pancreas is well vascularized.
4. Related to this, is it not conceivable bacteria translocate to pancreas upon DSS and stimulate GP2 expression directly, without a gut derived TNF α axis? Pancreatic bacterial load upon DSS should be quantified (Universal 16S Q-PCR per gram pancreas eg, or CFU on LB plate in mice with E.coli; E.coli GFP also ok).
5. From what age are WT and Gp2^{-/-} mice co-housed? Is the same difference in bacterial phyla found in littermates Wt vs pancreas-specific GP2^{-/-} mice if tamoxifen given early in life? The bacteria different just are typical ones to be different between strains of different origins and their housing histories and a causal role for Gp2 deficiency would need to be proven.

Minor:

- Title: "the first line of defense" should be changed to "a" first line of defense.
- Given the poor direct evidence of TNF α involvement as it stands I suggest taking it out of the abstract.

Reviewer #2 (Remarks to the Author):

General comments.

The authors have explored the role of the pancreatic glycoprotein 2 (GP2) to protect the epithelial barrier especially against the adherent-invasive Escherichia coli (AIEC). The aim of the authors is to provide solid data to show that GP2 could be a novel potential preventive and therapeutic tool to limit the level of AIEC and the disruption of the epithelial barrier observed in Inflammatory Bowel Disease patients.

They notably used DSS-induced colitis mice model in mice deficient in GP2 and they showed that

these mice were more inflamed with a higher mucosa-associated E. coli.

The experiments are well chosen and interpreted and the fact that GP2 is increased during intestinal inflammation is essential to further explore its role. They showed very clearly that the GP2 expression in the pancreas of inflamed mice is stimulated by an inflammatory signal as Tnf-alpha. This suggests the existence of a pancreatic-colon axis in the inflammatory conditions.

The authors have thus performed a complete study of the impact of GP2 on the adherence of AIEC and they also mentioned that serum GP2 is increased in IBD patients.

The potential translational impact of these findings is not clear in spite of convincing results and the authors should improve this aspect. They should mention that some molecules inhibiting the adhesion of AIEC through FimH are already in Phase Ib trial in IBD patients. The authors should mention the works performed by the group of Nicolas Barnich on this strategy for example through this citation: The potential of FimH as a novel therapeutic target for the treatment of Crohn's disease. Sivignon A(1), Bouckaert J(2), Bernard J(3)(4)(5), Gouin SG(6), Barnich N(1). *Expert Opin Ther Targets*. 2017 Sep;21(9):837-847. In this paper, the authors described their strategies to target these AIEC strains and to promote their intestinal clearance. They have developed a very nice anti-adhesive strategy that are based on the development of FimH antagonists. They discuss their potential as personalized microbiota-targeted treatments for CD patients abnormally colonized by AIEC.

The authors should evaluate the potential use of their findings in IBD patients. Would the GP2 delivery could be of interest in IBD patients? Would it be a good idea to combine their findings to already available inhibitors of AIEC adherence?

Specific comments.

1. The title could be more precise as in its actual state, it is a bit vague and it is difficult to guess against what the GP2 protein protects the mucus. The inflammation should probably be mentioned in a putative revised title.
2. The authors should precise why they are using mucosa-associated E. coli more than the abbreviation AIEC. AIEC is a more familiar concept than the more generalist mucosa-associated E. coli.
3. They showed the increase of GP2 and its protective role in the DSS-induced colitis model without justifying the choice of this model rather than the TNBS-induced colitis model.

Point-by-point responses to the reviewers' comments

Comments from reviewers are in bold italic letters.

Reviewer: 1

Reviewers' comments: Reviewer #1

(Remarks to the Author):

In their manuscript "Pancreatic Glycoprotein 2 as the First Line of Defense for Mucosal Protection", Kurashima et al. describe that pancreas and not intestine-derived GP-2 is the source of luminal GP-2 and helps "opsonize" bacteria. The benefit of bacterial binding only becomes evident upon intestinal barrier breakdown, in a model of DSS induced colitis. Conversely, barrier-breakdown also leads to increased pancreatic expression of GP-2, suggesting a communication from gut to pancreas. Though the concept of pancreas-derived antimicrobials is interesting, this is far from new for GP-2. The manuscript scratches only the surface of many aspects that could go into much more depth to answer the more interesting questions as to how e.g. auto-antibodies against GP-2 arise in IBDs and if they are at all relevant (are they found in the lumen? Do they really inhibit GP-2 function), what GP-2 may bind beyond FimH, if it is the only antimicrobial secreted by the exocrine pancreas, how the secretion is regulated on a granule level, what effect GP-2 has on the mucosal rather than luminal microbiome etc. Thus, the authors do not show much beyond what is known.

[Authors' response]

We thank the reviewer for their helpful comments and support to improve our manuscript. We have responded to the following key issues raised by the reviewer and conducted additional experiments to provide more data-based evidence and discussion:

1. *Auto-antibodies against GP-2 arise in IBDs and if they are at all relevant (are they found in the lumen? Do they really inhibit GP-2 function)*

[Authors' response]

Here, the reviewer has raised an important point regarding whether the autoantibodies to GP2 in inflammatory bowel disease (IBD) patients inhibit GP2. We agree that this an important issue; therefore, we conducted additional experiments to examine whether anti-GP2 autoantibodies are actually raised in the luminal contents of DSS-colitis mice. This also allowed us to carefully examine the hypothesis that infiltration of GP2-bound bacterial into the systemic compartment leads to the induction of luminal autoantibodies against GP2. To begin with, we immunized GP2-deficient mice with GP2 with complete Freund's adjuvant (CFA) as a positive

control, and found an increase of luminal anti-GP2 IgG in the GP2-immunized mice (revised Fig. 8h).

Next, we established two *in vivo* models. In the first, we administered dextran sodium sulfate (DSS) and examined whether these mice produced anti-GP2 autoantibodies. In the second, we systemically administered naïve mice with GP2-bound heat-killed *Escherichia coli* to produce a model of GP2-bound bacteria infiltrating from the gut to the systemic compartment and again examined the production of anti-GP2 autoantibodies. Both models showed significant induction of autoantibodies against GP2 (revised Fig. 8h). Taken together, these results provide the first experimental evidence that GP2-bound bacteria infiltrating into the systemic compartment induce autoantibodies against GP2 under an inflammatory milieu.

A recent publication has indicated that mice with severe *Salmonella typhi* infection show marked anti-GP2 antibody production (Derer S *et al.*, *Inflamm Bowel Dis* 2020). Based on our data showing an increase of systemic anti-*E. coli* antibodies in DSS-treated GP2-deficient mice (original Fig. 4i), we hypothesized that anti-GP2 autoantibodies in the luminal compartment regulate GP2 function. To examine this hypothesis, we conducted an *E. coli* binding assay using mouse luminal contents containing anti-GP2 autoantibodies and found that the luminal contents of intact mice had no inhibitory effect on GP2-binding to *E. coli*, whereas those of DSS-treated or of immunized with GP2 and CFA mice inhibited GP2-binding to *E. coli*. (revised Fig. 8g). In the revised manuscript, we present these new data on the inhibitory roles of anti-GP2 autoantibodies and their production in murine colitis together with mechanistic evidence (revised Fig. 8i and h), and we have expanded the discussion accordingly (lines 355–379 and 4502–514).

2. GP-2 may bind beyond FimH

[Authors' response]

To address this point, we developed FimH-deficient *E. coli* that were then exposed to human or murine GP2. Unlike wild-type *E. coli*, FimH-deficient *E. coli* had no reactivity against either of the GP2s, emphasizing that GP2 specifically binds to FimH. We have added these findings to the manuscript as revised Fig. 8c.

In addition, to examine the novel bacterial target of GP2, we isolated GP2-bound bacteria from the murine luminal contents by magnetic-activated cell sorting. Sequencing of the isolated bacterial colony revealed that *Lactobacillus* spp. (*L. johnsonii*, *L. intestinalis*, and *L. murinus*) and *Faecalibaculum rodentium* were selectively bound by GP2. It has been reported that systemic immune responses to the cell wall components of *L. johnsonii* (e.g., polysaccharides) are upregulated in the sera of IBD patients, suggesting that translocation of *L. johnsonii*, including

GP2-bound *L. johnsonii*, occurs via disruption of the mucosal barrier (Pasciak M *et al.*, *Microb Biotechnol* 2017). *Faecalibaculum rodentium* is a mucus bacteria found inside colon tumors (Zagato E *et al.*, *Nat Microbiol* 2020). Taken together, these findings show that GP2-mediated translocation of various bacterial species in normal gut environments is limited, whereas when the gut environment is disturbed, GP2-bound bacteria are able to translocate and induce GP2-specific autoantibodies. These findings are now shown in Supplemental Fig. 15 and discussed in the revised manuscript (lines 425–440).

3. Antimicrobial secreted by the exocrine pancreas

[Authors' response]

To address this point, we conducted an RNA-Seq analysis of pancreas isolated from intact and DSS-colitis mice and found that Reg (regenerating islet-derived) family antimicrobial proteins (i.e., Reg2, Reg3 α , Reg3 β , and Reg3 γ) were abundantly expressed and tended to be increased in the inflammatory condition, although this finding was not statistically significant. These results strongly support the assertion that the pancreas plays an important role as a protective organ during intestinal inflammation. We have added these data to the revised manuscript as revised Supplementary Fig. 7 and address our findings in the revised Discussion section (lines 174–181).

4. How the secretion is regulated on a granule level

[Authors' response]

We agree with the reviewer regarding the importance of the GP2 granule secretory system. We therefore examined the expression levels of genes involved in granular transport (*Rap1a*, *Rab3d*, *Rap1a*, *Rab8a*, *Rab27b*) and granular mobilization and fusion (*Vamp2*, *Vamp3*, *Vamp8*, *Stx3*, and *Stx7*) (Lee JS *et al* *Biophys Rep* 2018; Sabbatini ME *et al.*, *J Biol Chem* 2008; Kalus I *et al.*, *Biochem J* 2002; Yoon Y *et al.*, *J Cell Biol* 1998; and Marchelletta RR *et al.*, *Am J Physiol Cell Physiol* 2008). We found that none of these genes were upregulated in pancreatic acinar cells under inflammatory conditions, as shown in Supplementary Fig. 6a of the revised manuscript (lines 158–167).

Because GP2 is a glycosylphosphatidylinositol-anchored protein, we next examined whether glycosylphosphatidylinositol anchor cleavage was enhanced by an increase of enzyme expression via inflammatory signaling. We found that none of the genes encoding proteins involved in glycosylphosphatidylinositol anchor cleavage (e.g., carboxypeptidases, phospholipases, and trypsins) (Kim MJ *et al.*, *Pancreas* 2001; Hooper NM *et al.*, *Biochem J* 1997) were upregulated under the inflammatory condition. These data are now shown in revised Supplementary Fig. 6b (lines 167–173).

We found increased concentrations of GP2 in the pancreatic juice and luminal contents (Fig. 2e), which is consistent with our finding that inflammation increases the production of GP2 by acinar cells but does not increase granular release (mobilization and cleavage of GP2) (Supplementary Fig. 6). Collectively, we consider that increased production of GP2 by acinar cells with constitutive granular release leads to increased concentrations of luminal GP2 after inflammatory cytokine stimulation. We have added these data as revised Fig. 2e and Supplementary Fig. 6 in the revised manuscript and discuss our findings in the revised Discussion section (lines 152-157 and 405-419).

5. Mucosal rather than luminal microbiome

[Authors' response]

We showed in the original manuscript that wild-type mice and GP2-deficient mice had a similar microbiome profile, as determined by 16S rRNA analysis using fecal samples (original Fig. 4b). We also showed that the binding of GP2 to bacteria did not alter their proliferation (original Fig. 8b). Consequently, we concluded that GP2 has little effect on the diversity of the commensal microbiota and that it instead regulates the bacteria–host interaction to protect against invasion by FimH-expressing commensal and pathogenic bacteria (e.g., *E. coli* and *Salmonella* sp.).

To address the reviewer's comment, in addition to the luminal microbiota, we examined whether the mucosal microbiota is altered under conditions of GP2 deficiency. After removal of feces and luminal contents, we scraped the mucosal compartment to collect mucosal bacteria. Then, bacterial loads were determined to evaluate whether GP2 deficiency altered the number of bacteria in the mucosa with respect to the total bacterial load (*Eubacteria*) and to the sizes of several specific bacterial populations (e.g., *E. coli*, *Lactobacillus*, *Bacteroides*, *Helicobacter*, and *Proteus*). We detected no significant differences in the total bacterial load; however, the abundance of *E. coli* was significantly increased in GP2-deficient mice compared with in wild-type mice (revised Fig. 4c). Taking into account our data showing that GP2 is not essential for *E. coli* growth (original Fig. 8b), we considered that GP2 regulated the composition of the mucosal microbiota by controlling the translocation of bacteria to the mucosal compartment. We have added these new data as revised Fig. 4c and discuss our findings in lines 242–251.

Specific technical concerns:

1. Quantification by ELISA of GP-2 in pancreatic juice and in lumen along the gut in WT mice under homeostatic conditions and upon DSS would be more powerful than IHC of the lumen.

[Authors' response]

To address this issue, we developed a means of collecting pancreatic juice from mice via a catheter (See revised Materials and Methods section, lines 578–586; modification of Freedman SD et al., *Gastroenterology* 121, 950-957, 2001). Using this method, we found that the concentration of GP2 was significantly higher in the pancreatic juice and luminal contents of mice with gut inflammation compared with in that of intact mice (revised Fig. 2e).

2. Similarly WB against GP2 to show specificity of signal would be more reassuring- some staining in Gp2-/-mice seen (Fig. 4h, 6f)

[Authors' response]

To address this issue, we re-examined our original manuscript and noticed careless typos in the legend of original Fig. 4h. The red signal should have been labeled as EUB338 not GP2; we have corrected the legend in the revised manuscript. In original Fig. 6f, the caption was correct and so has not been changed in the revised manuscript. We apologize for these typos and appreciate the reviewer bringing this issue to our attention.

In addition to the monoclonal anti-mouse GP2 antibody (MBL International) that we used for the experiments described in the original manuscript, we have conducted additional experiments using a polyclonal antibody (Thermo Fisher Scientific) against GP2 and found GP2-specific signals in wild-type but not GP2-knockout mice. We have added these data as new Supplementary Fig. 3a. In addition, western blot analysis of GP2 confirmed the deletion of GP2 in GP2-deficient mice. This information has been added as new Supplementary Fig. 3b in the revised manuscript.

3. The TNFa dependence of GP-2 expression is not convincing: it is only based on 1.5 fold increase in vitro, and may not explain what is happening upon DSS (nor IBD patients without anti-TNF therapy, as TNF is more likely up); also secretion of GP-2 may be more important than expression. (see 1.). Can the authors show increased circulating/pancreatic TNFa upon DSS in their mice? Additionally, adding the serum of DSS treated versus controls onto the acinar cell cultures and looking at GP2 expression would be more convincing, to see if the TNFa levels, if raised at all, are sufficient. TNFa from creeping mesenteric fat is unlikely to reach acinar cells directly through the organ matrices, while the pancreas is well vascularized.

[Authors' response]

We felt that it was important to address this issue experimentally. To examine the involvement of tumor necrosis factor alpha (TNF- α) in GP2 expression, we conducted a series of additional experiments. First, we administered TNF- α -neutralizing or control antibodies to mice during the colitis induction period. Compared with control antibody administration, TNF- α -neutralizing antibody treatment reduced upregulation of the GP2 signal in the pancreas (revised Fig. 3j).

In addition, based on the reviewer's suggestion to determine the location of TNF- α production, we examined the concentration of TNF- α in the pancreas, omentum, mesenteric fat, retroperitoneal fat, epididymis fat, and circulation of DSS-colitis and control mice (new Fig. 3h). We found upregulation of TNF- α production in the blood circulation and the pancreas, as the reviewer suggested (revised Fig. 3i). We have added these new data as revised Figs. 3g–l and discuss our findings in lines 205–218.

To further address the issues raised by the reviewer, we also examined whether TNF- α administration upregulates GP2 expression and secretion into the pancreatic juice. We systemically administrated C57BL/6 intact mice with 100 ng TNF- α and found an increase of the GP2 signal and of the secretion of GP2. We have added these new data as revised Fig. 3e and f in the revised manuscript.

In addition, we also examined whether the upregulation of GP2 was relevant to patients with IBD by elucidating the concentration of GP2 in the stools of patients with IBD. We found significant upregulation of GP2, particularly in patients with Crohn's disease. These data are now shown as Fig. 2f of the revised manuscript.

4. Related to this, is it not conceivable bacteria translocate to pancreas upon DSS and stimulate GP2 expression directly, without a gut derived TNF α axis? Pancreatic bacterial load upon DSS should be quantified (Universal 16S Q-PCR per gram pancreas eg, or CFU on LB plate in mice with E.coli; E.coli GFP also ok).

[Authors' response]

To address the reviewer's concerns, we conducted additional experiments to examine bacterial translocation to the pancreas. We collected pancreas, and mesenteric lymph nodes as a positive control, and examined bacterial translocation (as assessed by culture of samples on LB and BHI plates). We found bacterial translocation into the mesenteric lymph nodes but not into the pancreas (revised Supplementary Fig. 9), indicating that translocation of bacteria to the pancreas is not a direct stimulus for GP2 release.

5. From what age are WT and Gp2^{-/-}-mice co-housed? Is the same difference in bacterial phyla found in littermates Wt vs pancreas-specific GP2^{-/-} mice if tamoxifen given early in life? The bacteria different just are typical ones to be different between strains of different origins and their housing histories and a causal role for Gp2 deficiency would need to be proven.

[Authors' response]

We agree that describing the housing history is important for understanding the luminal microbiota. As we showed in the original manuscript, the bacterial composition was unchanged between wild-type and GP2-deficient mice at the phylum and genus levels (original Fig. 4b and Supplementary Fig. 10a). Our experiments were conducted using littermate mice that had been co-housed from 2 weeks of age; we have added this information to the Materials and Methods section of the revised manuscript (line 544).

To examine whether the same difference in bacterial phyla is present in littermates of wild-type and pancreas-specific GP2^{-/-} mice, we examined the microbiota of two-week-old mice treated with tamoxifen (see revised Supplementary Fig. 11a and lines 548–550 of the revised manuscript for the experimental protocol). We did not find any obvious differences between the bacterial populations of the tamoxifen-treated mice (revised Supplementary Fig. 11b). These results indicated that GP2-deficiency did not influence the diversity of the luminal microbiota. We have added these new data as revised Supplementary Fig. 11 and discuss our findings in the revised manuscript (lines 229–241).

Minor:

-Title: "the first line of defense" should be changed to "a" first line of defense.

[Authors' response]

We have amended the title of our manuscript as suggested.

-Given the poor direct evidence of TNF α involvement as it stands I suggest taking it out of the abstract.

[Authors' response]

We understand the reviewer's concern based on the data presented in our previous manuscript; however, after conducting the additional experiments to address the reviewers' comments (e.g., specific technical concern #4), we now provide additional supportive evidence for the importance of TNF- α in the regulation of pancreatic GP2 (Fig. 3e–j). Therefore, we would

like to keep the statement regarding the involvement of TNF- α in the Abstract. To also address the reviewer's concerns related to human situation by Reviewer#2, we now mention the increase of GP2 in Crohn's disease patients in the revised abstract.

Reviewer #2

(Remarks to the Author):

General comments. The authors have explored the role of the pancreatic glycoprotein 2 (GP2) to protect the epithelial barrier especially against the adherent-invasive Escherichia coli (AIEC). The aim of the authors is to provide solid data to show that GP2 could be a novel potential preventive and therapeutic tool to limit the level of AIEC and the disruption of the epithelial barrier observed in Inflammatory Bowel Disease patients. They notably used DSS-induced colitis mice model in mice deficient in GP2 and they showed that these mice were more inflamed with a higher mucosa-associated E. coli. The experiments are well chosen and interpreted and the fact that GP2 is increased during intestinal inflammation is essential to further explore its role. They showed very clearly that the GP2 expression in the pancreas of inflamed mice is stimulated by an inflammatory signal as Tnf-alpha. This suggests the existence of a pancreatic-colon axis in the inflammatory conditions. The authors have thus performed a complete study of the impact of GP2 on the adherence of AIEC and they also mentioned that serum GP2 is increased in IBD patients. The potential translational impact of these findings is not clear in spite of convincing results and the authors should improve this aspect. They should mention that some molecules inhibiting the adhesion of AIEC through FimH are already in Phase Ib trial in IBD patients. The authors should mention the works performed by the group of Nicolas Barnich on this strategy for example through this citation: The potential of FimH as a novel therapeutic target for the treatment of Crohn's disease. Sivignon A(1), Bouckaert J(2), Bernard J(3)(4)(5), Gouin SG(6), Barnich N(1). Expert Opin Ther Targets. 2017 Sep;21(9):837-847. In this paper, the authors described their strategies to target these AIEC strains and to promote their intestinal clearance. They have developed a very nice anti-adhesive strategies that are based on the development of FimH antagonists. They discuss their potential as personalized microbiota-targeted treatments for CD patients abnormally colonized by AIEC. The authors should evaluate the potential use of their findings in IBD patients. Would the GP2 delivery could be of interest in IBD patients? Would it be a good idea to combine their findings to already available inhibitors of AIEC adherence?

[Authors' response]

We thank the reviewer for their helpful and supportive comments. We are also pleased to learn that the referee recognizes the importance of our study and thinks that it has a solid experimental design. The reviewer raised several important points to improve our manuscript, and we have conducted additional experiments to address the issues raised.

An issue for the future clinical translation of the present observations is that we first have to perform experiments using human recombinant GP2 (rGP2) and compare the findings with the murine data. To begin to address this issue, we conducted experiments with *E. coli* with and without FimH and found that human rGP2 bound specifically to FimH (revised Fig. 8c). This finding is consistent with the data generated by using murine rGP2 and implies that human rGP2 has a similar immunobiological function to murine rGP2. We discuss these new data in the revised manuscript (lines 345–347 and 459–464).

We agree with the reviewer that it is important to consider the clinical relevance of our study and how to advance our findings to the clinical setting for the control of IBD. To address this point, we measured GP2 concentration in the stools of patients with IBD and obtained data that agree with the mouse colitis data, that is, GP2 concentration was significantly increased in Crohn's disease. These findings are now included as revised Fig. 2f and are discussed in lines 154–157 of the revised manuscript.

In terms of FimH inhibitors [e.g., monovalent heptyl mannose derivatives, monovalent thiazolyl amino-mannosides, *n*-heptyl α -D-mannose-based glycopolymers, as discussed in the review article by Sivignon et al. in *Expert Opinion on Therapeutic Targets* (2017)], all of the inhibitors so far reported remain the properties of their respective laboratories. Therefore, we are currently unable to directly examine the efficacy of combination of FimH antagonists on the possible control of GP2-mediated inflammation. However, to begin advancing our findings to the clinical setting, we prepared a large amount of recombinant mouse GP2 (rGP2) and conducted *in vivo* experiments to examine whether oral administration of rGP2 can control murine gut inflammation. As shown in Appendix Fig. 1, three oral inoculations of 100 μ g of rGP2 in a murine colitis model (DSS colitis) did not show a statistically significant improvement of body weight (an example of wasting symptom associated with gut inflammation) or of colon shortening (an inflammatory signature) in mice treated with oral rGP2, possibly because of the amount, delivery formulation, and/or route of rGP2 administration (e.g., oral vs. systemic vs. rectal).

We appreciate the reviewer's opinion of the clinical relevance of our study, and we certainly agree with the point that enhancing GP2 production in the pancreas may be a novel strategy for regulating intestinal inflammation. We feel that addressing whether the use of GP2 is a new

therapeutic target (or not) is beyond the scope of the present study, because our study's primary focus was to demonstrate the role of the pancreas as a novel organ associated with the GP2-mediated pancreas–gut axis and its involvement in the regulation of gut homeostasis and inflammation. However, we do address the clinical application of our findings in the revised manuscript (lines 459–464 and 476–494).

Specific comments.

1. The title could be more precise as in its actual state, it is a bit vague and it is difficult to guess against what the GP2 protein protects the mucus. The inflammation should probably be mentioned in a putative revised title.

[Authors' response]

We appreciate the suggestion regarding improving our title. We have changed the title to “Pancreatic Glycoprotein 2 as a First Line of Defense for Mucosal Protection in Intestinal Inflammation” in the revised manuscript.

2. The authors should precise why they are using mucosa-associated E. coli more than the abbreviation AIEC. AIEC is a more familiar concept than the more generalist mucosa-associated E. coli.

[Authors' response]

In accordance with the reviewer's comment, we have changed mucosa-associated *E. coli* to adherent–invasive *E. coli* in the revised manuscript.

3. They showed the increase of GP2 and its protective role in the DSS-induced colitis model without justifying the choice of this model rather than the TNBS-induced colitis model.

[Authors' response]

We felt that it was important to address the issue experimentally; therefore, we conducted an experiment using a 2,4,6-trinitrobenzene sulfonic acid (TNBS)-induced colitis model. When wild-type and GP2-deficient mice were exposed to TNBS to induce colitis, we found that pancreatic GP2 was increased in the wild-type mice with TNBS-colitis (Supplementary Fig. 12d). Importantly, GP2-deficient mice showed more severe intestinal inflammation with bacterial translocation and inflammatory cell infiltration compared with that in the wild-type mice (revised Fig. 4j and k). We have added these new results as revised Fig. 4 and Supplementary Fig. 12 in the revised manuscript and discuss our findings in lines 266–277.

REVIEWERS' COMMENTS

Reviewer #1 (Remarks to the Author):

The authors did an impressive body of work to address all concerns raised and I believe the manuscript is sound as stands! The only thing I would recommend discussing is the issue of the best therapeutic strategy: On the one hand elevating luminal GP2 could be beneficial, on the other hand the very patients one may want to deliver GP2 as a therapeutic to may already have neutralizing autoantibodies. Also, if there is pre-existing immunity against GP2, giving more GP2, even if luminary, may act as a "booster" and actually be detrimental. So make clear the therapy may only work in GP2 Ab-negative patients. Overall I think the focus on therapy can be dimmed down-the mechanism is interesting enough now.

Reviewer #2 (Remarks to the Author):

The authors have shown the role of the pancreas as a novel organ associated with the GP-2 mediated pancreas-gut axis. In order to address the question on the future clinical translation of their observations, they have started to work with human recombinant GP2 (rGP2). They obtained promising preliminary data with rGP2 which has a similar immunomodulatory function to murine GP2. They have also measured GP2 in the stools of IBD patients and have confirmed the increase of GP2 observed in mice.

They have taken into account th recommendation to include the term inflammation in their new title and to use the term of AIEC.

Finally, they have tested the protective role of GP2 in a TNBS-induced colitis model and and they have shown that GP2 was increased in the inflamed wild type mice.

Point-by-point responses to the reviewers' comments

NCOMMS-19-14995C

Comments from reviewers are in bold italic letters.

REVIEWERS' COMMENTS

Reviewer #1 (Remarks to the Author):

The authors did an impressive body of work to address all concerns raised and I believe the manuscript is sound as stands! The only thing I would recommend discussing is the issue of the best therapeutic strategy: On the one hand elevating luminal GP2 could be beneficial, on the other hand the very patients one may want to deliver GP2 as a therapeutic to may already have neutralizing autoantibodies. Also, if there is pre-existing immunity against GP2, giving more GP2, even if luminary, may act as a "booster" and actually be detrimental. So make clear the therapy may only work in GP2 Ab-negative patients. Overall I think the focus on therapy can be dimmed down-the mechanism is interesting enough now.

We thank to Reviewer #1 for recognizing our efforts in responding to the reviewer's constructive and helpful comments. According to the recommendation of the Reviewer #1, we have modified our discussion on the therapeutic application of GP2 as described below:

"Elevating luminal GP2 could be beneficial; on the other hand, the very patients one may want to deliver GP2 as a therapeutic to may already have neutralizing autoantibodies. Also, if there is pre-existing immunity against GP2, administration of additional GP2, even if luminary, may act as a "booster" and actually be detrimental. Therefore, it is forethoughtfully examined the GP2 therapy to the GP2 autoantibodies-negative patients." in the revised manuscripts (line 466-470).

Reviewer #2 (Remarks to the Author):

The authors have shown the role of the pancreas as a novel organ associated with the GP-2 mediated pancreas-gut axis. In order to address the question on the future clinical translation of their observations, they have started to work with human recombinant GP2 (rGP2). They obtained promising preliminary data with rGP2 which has a similar immunomodulatory function to murine GP2. They have also measured GP2 in the stools of IBD patients and have confirmed the increase of GP2 observed in mice.

They have taken into account th recommendation to include the term inflammation in their new title and to use the term of AIEC.

Finally, they have tested the protective role of GP2 in a TNBS-induced colitis model and and they have shown that GP2 was increased in the inflamed wild type mice.

We thank to Reviewer #2 for the recognition of the importance of our study.